# Equal-time approach to real-time dynamics of quantum fields

Robert Ott[1], Torsten V. Zache[2,3] and Jürgen Berges[1]

**1** Heidelberg University, Institut für Theoretische Physik,
Philosophenweg 16, 69120 Heidelberg, Germany
**2** Institute for Theoretical Physics, University of Innsbruck, 6020 Innsbruck, Austria
**3** Institute for Quantum Optics and Quantum Information
of the Austrian Academy of Sciences, 6020 Innsbruck, Austria

## Abstract

We employ the equal-time formulation of quantum field theory to derive effective kinetic theories, first for a weakly coupled non-relativistic Bose gas, and then for a strongly correlated system of self-interacting $N$-component fields. Our results provide the link between state-of-the-art measurements of equal-time effective actions using quantum simulator platforms, as employed in Refs. [1, 2], and observables underlying effective kinetic or hydrodynamic descriptions. New non-perturbative approximation schemes can be developed and certified this way, where the a priori time-local formulation of the equal-time effective action has crucial advantages over the conventional closed-time-path approach which is non-local in time.



# 1  Introduction

Quantum fields describe the microphysical laws of nature and are relevant for quantum technology when devices become large. Despite their relevance, the complex dynamical properties of quantum fields are to a large extent unknown because ab initio simulations in real time are in general beyond capabilities of classical computers. Quantum simulators open up a way forward, and prominent examples with ultra-cold atoms include the simulation of relaxation and (pre-)thermalization dynamics [3–6], many-body localization [7,8], quantum scars [9–11], and universal dynamics far from equilibrium [12–14].

Ultra-cold atom measurements are typically done at snapshots in time with the important ability to extract equal-time correlations to high orders [15,16]. Equal-time correlations are highly suitable for the description of non-equilibrium systems, similar in spirit – but not limited to – kinetic descriptions in terms of single-time distributions. However, in contrast to these time-local approaches the conventional formulation of non-equilibrium quantum field theory is based on the closed-time-path contour [17–19] involving multiple-time correlations, which are difficult to access experimentally. In particular, standard derivations of effective kinetic descriptions from quantum field theory start from non-local equations in time which become time-local only after a series of approximations [20,21].

In this work we derive effective kinetic theories for an ultra-cold Bose gas starting from an equal-time formulation of quantum field theory [1,2,22]. The central quantity is the time-dependent quantum effective action $\Gamma_t$, which contains the same information as the density operator at time $t$, but is expressed in terms of equal-time correlations. From the functional evolution equation for $\Gamma_t$ [22] we derive evolution equations for equal-time vertices, which may be directly extracted from quantum simulation results as pioneered in Refs. [1,2]. Here we demonstrate that the two- and four-point correlation functions at equal times contain the complete information for the derivation of the Boltzmann equation for a weakly coupled non-relativistic Bose gas, and of an effective kinetic theory for a strongly correlated system of self-interacting $N$-component fields [23]. Our results establish a direct link between equal-time correlations and observables underlying effective kinetic or hydrodynamic descriptions. The approach thus opens up new possibilities to develop and certify novel approximation schemes for the dynamics of complex quantum many-body systems.

## 2  Model and overview

We consider an $N$-component non-relativistic scalar field theory with Hamiltonian

$$\hat{H} = \int_x \left[ \frac{\nabla \hat{\Psi}_x^\dagger \nabla \hat{\Psi}_x}{2m} - \mu \hat{\Psi}_x^\dagger \hat{\Psi}_x + \frac{g}{4N} : (\hat{\Psi}_x^\dagger \hat{\Psi}_x)^2 : \right]. \tag{1}$$

Here $\hat{\Psi}_x = (\hat{\psi}_{x,1}, ..., \hat{\psi}_{x,N})$ is the $N$-component field operator at spatial position $x$, $g$ is the scattering constant, $m$ denotes the mass of the atoms, $\mu$ represents the chemical potential and the colons indicate normal ordering of operators. The field operators fulfill canonical commutation relations $[\hat{\psi}_{x,i}, \hat{\psi}_{y,j}^\dagger] = \delta(x-y)\delta_{ij}$, where we employ natural units with setting $\hbar = 1$. Here and in the following, we use short-hand notations for integrals over spatial coordinates $\int_x = \int_{-\infty}^{\infty} \mathrm{d}^3 x$ and momenta $\int_k = \int_{-\infty}^{\infty} \mathrm{d}^3 k/(2\pi)^3$. We focus on three spatial dimensions where Eq. (1) may be considered as a low-energy effective theory for ultra-cold Bose gases with a U($N$) symmetry.

We define a generating functional for equal-time correlations as [22]

$$Z_t[\mathbf{J}^{(*)}] = \mathrm{Tr}\left( \hat{\rho}_t e^{\int_x (\hat{\Psi}_x^\dagger \mathbf{J}_x + \mathbf{J}_x^* \hat{\Psi}_x)} \right), \tag{2}$$

where $\hat{\rho}_t$ denotes the time-dependent density operator and $\mathbf{J}_x^{(*)} = (J_{x,1}^{(*)}, ..., J_{x,N}^{(*)})$ are the $N$-component source fields. The generating functional contains the same information as the $t$-dependent density operator and fully describes the underlying quantum system at time $t$. With this representation the system is completely characterized by its set of equal-time correlations and its evolution is determined by the Hamiltonian of the theory. Repeated differentiation with respect to the sources, and evaluation for vanishing sources, yields symmetrically ordered correlation functions

$$G^{(n)}_{\alpha_1..\alpha_j, \alpha_{j+1}..\alpha_n}(t) = \frac{1}{Z_t[\mathbf{J}^{(*)}]} \frac{\delta}{\delta J_{\alpha_1}^*} \cdots \frac{\delta}{\delta J_{\alpha_j}^*} \frac{\delta}{\delta J_{\alpha_{j+1}}} \cdots \frac{\delta}{\delta J_{\alpha_n}} Z_t[\mathbf{J}^{(*)}] \Big|_{\mathbf{J}, \mathbf{J}^*=0}, \tag{3}$$

where we abbreviated the spatial and component indices as $\alpha_i$, e.g. $\alpha_1 = (x_1, i_1)$. According to (3) we associate fields $\hat{\psi}$ with the indices to the left (here $\alpha_1 \cdots \alpha_j$), and conjugate fields $\hat{\psi}^\dagger$ with the rightmost indices ($\alpha_{j+1} \cdots \alpha_n$). Throughout this work we consider the case of U($N$) invariant correlations in the non-relativistic theory. By choosing a U($N$) invariant initial state the symmetry is preserved for the dynamics with Hamiltonian (1). As a consequence, all non-vanishing correlation functions involve an equal number of field and conjugate field operators. Specifically, this yields one type of two-point function which is given by

$$G^{(2)}_{\alpha_1, \alpha_2}(t) = \frac{1}{2} \langle \{\hat{\psi}_{x_1, i_1}, \hat{\psi}_{x_2, i_2}^\dagger\} \rangle_t, \tag{4}$$

where $\{\cdot, \cdot\}$ denotes the anti-commutator of operators, and the expectation value is given by the trace with respect to the density operator at time $t$.

In general, we distinguish between connected and disconnected correlation functions. Connected correlation functions (superscript "$\mathbf{c}$") are obtained by differentiating with respect to the equal-time Schwinger functional $W_t = \log(Z_t)$,

$$G^{\mathbf{c},(n)}_{\alpha_1..\alpha_j, \alpha_{j+1}..\alpha_n}(t) = \frac{\delta}{\delta J_{\alpha_1}^*} \cdots \frac{\delta}{\delta J_{\alpha_j}^*} \frac{\delta}{\delta J_{\alpha_{j+1}}} \cdots \frac{\delta}{\delta J_{\alpha_n}} W_t[\mathbf{J}^{(*)}] \Big|_{\mathbf{J}, \mathbf{J}^*=0}. \tag{5}$$

At order $2n$, they contain information about correlations of $n$ bodies. Conversely, $n$-th order *disconnected* correlation functions are given by sums of all combinations of *connected* correlations involving in total $n/2$ bodies. For example, for $n = 4$ one gets (for the U($N$)-invariant case)

$$G^{(4)}_{\alpha_1 \alpha_2, \alpha_3 \alpha_4} = G^{\mathbf{c},(4)}_{\alpha_1 \alpha_2, \alpha_3 \alpha_4} + G^{\mathbf{c},(2)}_{\alpha_1, \alpha_3} G^{\mathbf{c},(2)}_{\alpha_2, \alpha_4} + G^{\mathbf{c},(2)}_{\alpha_1, \alpha_4} G^{\mathbf{c},(2)}_{\alpha_2, \alpha_3}, \tag{6}$$

and for $n = 2$ we have $G^{\mathbf{c},(2)}_{\alpha_1,\alpha_2} = G^{(2)}_{\alpha_1,\alpha_2}$ since the one-point function vanishes.

The dynamics of quantum fields is often addressed in terms of effective kinetic theories for a time-dependent distribution function $f_p(t)$, where we consider the case of spatially tanslation invariant systems to ease the notation and the momentum $p$ is obtained from Fourier transformation with respect to relative coordinates. In the following, using U($N$) symmetry the distribution function is obtained without loss of generality from the diagonal two-point correlation function $G^{(2)}_{\alpha_1,\alpha_2} \to G^{(2)}_{x_1,x_2} \delta_{i_1,i_2}$ in Fourier space as

$$G^{(2)}_p(t) = f_p(t) + \frac{1}{2} . \tag{7}$$

Starting from an exact evolution equation for the time-dependent quantum effective action obtained as the Legendre transform of $W_t$ in section 3, we will derive effective kinetic equations of the form

$$\partial_t f_p(t) = \int_{q,r,s} |T_{pqrs}(t)|^2 \big( (f_p(t) + 1)(f_q(t) + 1)f_r(t)f_s(t) - f_p(t)f_q(t)(f_r(t) + 1)(f_s(t) + 1) \big) . \tag{8}$$

It shows characteristic "gain" and "loss" terms describing (in this case) $2 \leftrightarrow 2$ scattering into and out of the momentum mode $p$.

We first compute the dynamics of the Bose gas using a perturbative expansion in the small interaction strength $g \ll 1$ in section 4, where (8) reduces to the Boltzmann equation describing a dilute medium with occupancy $f_p \ll \mathcal{O}(1/g)$ such that particles stream freely in between individual scatterings. In this simplest case one finds from the (irreducible part) of the equal-time four-point function $G^{\mathbf{c},(4)}$ a time- and momentum-independent matrix element $|T_{pqrs}(t)|^2 = g^2/2(2\pi)^3\delta(p + q - r - s)(2\pi)\delta(\Delta\omega_{pqrs})$, where $\Delta\omega_{pqrs} = \omega_p + \omega_q - \omega_r - \omega_s$ is the single-particle energy difference of in- and out-going particles. Therefore, one recovers that the scattering rate is given by the asymptotic T-matrix elements $|T_{pqrs}|^2$ in vacuum in this case.

In section 5 a non-perturbative approximation scheme is considered, where we employ an expansion in the number of field components $N$. At next-to-leading order in the large-$N$ expansion we again recover an effective kinetic equation of the form (8), however, in this case with a time- and momentum-dependent $|T_{pqrs}(t)|^2$. We demonstrate that the latter is also fully determined by the irreducible part of the equal-time four-point function, which implements a geometric series resummation of the distribution $f_p(t)$ itself such that one obtains a closed equation for the time evolution. The importance of the large-$N$ kinetic theory is that it can describe also strongly correlated systems with non-perturbatively high occupancies [24].

While these results establish a direct link between equal-time correlations and typical observables underlying effective kinetic theories, the exact quantum evolution equations we derive from the equal-time effective action are not limited to kinetic theory approximations. In section 6 we discuss an experimental protocol of how the exact equations could be established in quantum simulations with ultra-cold atom platforms, extending the procedures of Refs. [1,2] to the Bose fields appearing in the defining Hamiltonian.

## 3 Equal-time 1PI effective action

In this section, we introduce the equal-time effective action and the corresponding time-dependent vertices, which are the irreducible building blocks of all connected equal-time correlation functions. The equal-time one-particle irreducible (1PI) effective action [1, 2, 22],

$$G^{\mathbf{c},(6)} = \quad \text{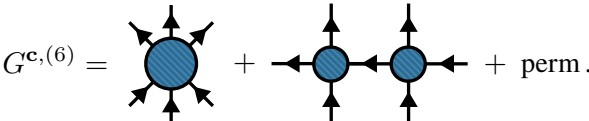} \quad + \text{perm}.$$

Figure 1: Connected correlation functions from irreducible building blocks. We show the diagrammatic contributions to the 1PI connected six-point function ($G^{\mathbf{c},(6)}$). The first term involves a 1PI six-vertex, the second is assembled from two four-vertices. Field indices and permutations of external legs are implied and the number of in- and outgoing arrows is conserved due to U($N$) invariance. For details, see also appendix A.

analogous to the free energy, is defined as the Legendre transform

$$\Gamma_t[\Psi^{(*)}] = -W_t[\mathbf{J}^{(*)}] + \int_x \left( \Psi_x^\dagger \mathbf{J}_x + \mathbf{J}_x^* \Psi_x \right), \tag{9}$$

with field-dependent sources $\mathbf{J}(\Psi)$, $\mathbf{J}^*(\Psi^*)$, and $\Psi_x^{(*)}(\mathbf{J}^{(*)}) = \langle \hat{\Psi}_x^{(\dagger)} \rangle_J$, where the expectation value is defined with respect to the trace in Eq. (2) in the presence of sources. The effective action can be expanded in terms of the fields as

$$\Gamma_t[\Psi^{(*)}] = \sum_{n=2}^{\infty} \Gamma^{(n)}_{x_1...x_n,i_1...i_n}(t) \times \psi^*_{x_1,i_1}...\psi_{x_n,i_n}, \tag{10}$$

with 1PI equal-time vertices that are obtained by differentiation as

$$\Gamma^{(n)}_{x_1...x_n,i_1...i_n}(t) = \left. \frac{\delta^n \Gamma_t}{\delta \psi^*_{x_1,i_1}...\delta \psi_{x_n,i_n}} \right|_{\Psi^*,\Psi=0}. \tag{11}$$

These 1PI vertices are the irreducible building blocks for connected correlation functions. Specifically, this means that any equal-time connected correlation function is a combination of equal-time vertices and two-point functions. Important relations of correlation functions and effective vertices can be obtained by the definitions of the Schwinger functional and effective action, in combination with the chain rule for derivatives with respect to fields and sources. As an example, one finds for the two-point functions the relation $G^{\mathbf{c},(2)}_{\alpha_1\alpha_2} = (\Gamma^{(2)})^{-1}_{\alpha_1\alpha_2}$, and for four-point functions

$$G^{\mathbf{c},(4)}_{\alpha_1\alpha_2,\alpha_3\alpha_4} = -G^{\mathbf{c},(2)}_{\alpha_1\alpha_1'} G^{\mathbf{c},(2)}_{\alpha_2\alpha_2'} G^{\mathbf{c},(2)}_{\alpha_3\alpha_3'} G^{\mathbf{c},(2)}_{\alpha_4\alpha_4'} \Gamma^{(4)}_{\alpha_1'\alpha_2'\alpha_3'\alpha_4'}. \tag{12}$$

Here, every index $\alpha_i'$ of the four-vertex is contracted with an equal-time two-point function also carrying a corresponding external index $\alpha_i$. To illustrate this relation graphically, we introduce the following notation

$$G^{\mathbf{c},(2)}_{\alpha_1\alpha_2} \equiv \quad \longleftarrow \quad , \tag{13}$$

$$\Gamma^{(2)}_{\alpha_1\alpha_2} = \left( G^{\mathbf{c},(2)} \right)^{-1}_{\alpha_1\alpha_2} \equiv \quad \cdots\!\blacktriangleleft\!\cdots \quad , \tag{14}$$

$$\Gamma^{(n)}_{\alpha_1..\alpha_n} \equiv \quad , \tag{15}$$

where each ending line represents an index $\alpha$, and 1PI vertices are amputated (red bars). Lines can furthermore meet at bare vertices, which in the evolution equations arise in combination with either one or three inverse two-point functions attached to them, as we will establish shortly.

In general, diagrams are assembled by connecting lines with vertices, which is accompanied by integration and summation over vertex positions and component indices as summarized by $\alpha$. Importantly, the arrow indicates the flow of particles, such that all diagrams should conserve the arrows along the attached lines. While this diagrammatic language is very similar to perturbation theory in conventional quantum field theory, the equal-time correlation functions here depend on spatial coordinates and an overall time argument, rather than a set of spacetime coordinates, and no time-integrals appear.

Assuming spatial translation invariance, it will be beneficial to transform these objects and their evolution equations to Fourier space, where two-point functions and vertices are assigned momentum variables. Overall, momentum variables are assigned in a momentum conserving manner, i.e. a $\delta$-distribution $(2\pi)^3\delta(p_1 + p_2 - p_3 - p_4)$ is implied at each vertex with ingoing momenta $p_1, p_2$ and outgoing ones $p_3, p_4$. While connected four-point correlations originate from the four-vertex, all connected $n$-point correlations with $n > 4$ are built from sums over different diagrams involving vertices $\Gamma^{(4)}, .., \Gamma^{(n)}$. For example, the diagrams corresponding to the connected six-point function are displayed in Fig. 1. To clarify the diagrammatic rules, explicit formulae for diagrams are given in appendix A.

In the following, we first consider $N = 1$. The straightforward generalization to the $N$ component field theory will become important later for non-perturbative approximations based on an expansion in powers of $1/N$.

The effective action obeys an exact flow equation [22], which for the current model reads

$$i\partial_t \Gamma_t = \int_x \left( \frac{\delta\Gamma_t}{\delta\psi_x}\left(\frac{\nabla^2}{2m} + \mu\right)\psi_x - \frac{\delta\Gamma_t}{\delta\psi_x^*}\left(\frac{\nabla^2}{2m} + \mu\right)\psi_x^* \right. \tag{16}$$

$$\left. + \frac{g/2}{Z_t[J^{(*)}]}\frac{\delta^3 Z_t[J^{(*)}]}{(\delta J_x)^2 \delta J_x^*}\frac{\delta\Gamma_t}{\delta\psi_x^*} - \frac{g/2}{Z_t[J^{(*)}]}\frac{\delta^3 Z_t[J^{(*)}]}{\delta J_x(\delta J_x^*)^2}\frac{\delta\Gamma_t}{\delta\psi_x} - \frac{g}{8}\psi_x^*\frac{\delta\Gamma_t}{\delta\psi_x}\frac{\delta\Gamma_t}{\delta\psi_x^*}\frac{\delta\Gamma_t}{\delta\psi_x^*} + \frac{g}{8}\frac{\delta\Gamma_t}{\delta\psi_x}\frac{\delta\Gamma_t}{\delta\psi_x}\frac{\delta\Gamma_t}{\delta\psi_x^*}\psi_x \right),$$

where $J^{(*)} = J^{(*)}[\psi^{(*)}]$, such that the effective action is a functional of the fields $\psi^{(*)}$ [25]. Similar to the von-Neumann equation for the density operator, the evolution equation of the equal-time effective action is time-local. This is different from functional approaches involving unequal-time effective actions, where the system's history enters at each step of the evolution. The first two lines of Eq. (16) represent the terms also present in the classical-statistical theory, while the third line represents genuine quantum corrections. The term $\sim \delta^3 Z_t[J^{(*)}]/((\delta J_x)^2 \delta J_x^*)$ corresponds to a symmetrized third-order correlation function which may be written in terms of the effective action. To this end, we split it into connected and disconnected correlations

$$\langle\hat{\psi}_x^\dagger\hat{\psi}_x^\dagger\hat{\psi}_x\rangle_{\text{sym}} = \langle\hat{\psi}_x^\dagger\hat{\psi}_x^\dagger\hat{\psi}_x\rangle_{\text{sym}}^{\mathbf{c}} + 2G_{xx}^{\mathbf{c},(2)}\psi_x^* + \langle\hat{\psi}_x^\dagger\hat{\psi}_x^\dagger\rangle^{\mathbf{c}}\psi_x + (\psi_x^*)^2\psi_x, \tag{17}$$

where "sym" implies the symmetrization over all operator orderings. We furthermore have $G_{xx}^{\mathbf{c},(2)} = (\Gamma^{(2)})_{xx}^{-1}$ and $\langle\hat{\psi}_x^\dagger\hat{\psi}_x^\dagger\hat{\psi}_x\rangle_{\text{sym.}}^{\mathbf{c}} = -(\Gamma^{(2)})_{xy_1}^{-1}(\Gamma^{(2)})_{xy_2}^{-1}(\Gamma^{(2)})_{y_3x}^{-1}\Gamma^{(3)}_{y_1y_2y_3}$. Since odd orders of correlations vanish in the absence of a mean field $(\psi^{(*)}[J^{(*)} = 0] = 0)$, these contributions only contribute in the presence of further field derivatives.

Differentiation with respect to the fields $\psi^{(*)}$ yields the evolution equations for the inverse propagators and vertices. After applying the derivatives $\delta^2/\delta\psi_x^*\delta\psi_y$, and evaluating

the resulting expression for $\psi^{(*)}[J^{(*)} = 0] = 0$, we get

$$i\partial_t \Gamma_{xy}^{(2)} = \left( \frac{\nabla_y^2}{2m} - \frac{\nabla_x^2}{2m} + g(\Gamma_{xx}^{(2)})^{-1} - g(\Gamma_{yy}^{(2)})^{-1} \right) \Gamma_{xy}^{(2)}$$
$$+ \frac{g}{2} \int_{\mathbf{y}} \Gamma_{xy_4}^{(2)}(\Gamma_{y_4 y_1}^{(2)})^{-1}(\Gamma_{y_4 y_2}^{(2)})^{-1}(\Gamma_{y_3 y_4}^{(2)})^{-1}\Gamma_{y_1 y_2 y_3 y}^{(4)}$$
$$- \frac{g}{2} \int_{\mathbf{y}} \Gamma_{xy_3 y_1 y_2}^{(4)}(\Gamma_{y_1 y_4}^{(2)})^{-1}(\Gamma_{y_2 y_4}^{(2)})^{-1}(\Gamma_{y_4 y_3}^{(2)})^{-1}\Gamma_{y_4 y}^{(2)}, \tag{18}$$

where $\mathbf{y}$ refers to the set of integration variables $y_1, .., y_4$. For translationally invariant systems we switch to Fourier space, where the expression simplifies to

$$i\partial_t \Gamma_p^{(2)} = -\frac{g}{2} \int_{q,r,s} \Gamma_p^{(2)}(\Gamma_q^{(2)})^{-1}(\Gamma_r^{(2)})^{-1}(\Gamma_s^{(2)})^{-1}(\Gamma_{pqrs}^{(4)} - \Gamma_{rspq}^{(4)}). \tag{19}$$

Here, we used the definition $\Gamma_p^{(2)}(2\pi)^3 \delta(p-q) = \int_{xy} \exp(ipx - iqy)\Gamma_{xy}^{(2)}$, see also corresponding expressions in section A. Similarly, the four-vertices $\Gamma_{pqrs}^{(4)}$ carry a momentum conserving delta distribution $(2\pi)^3 \delta(p+q-r-s)$ which will be implied throughout the rest of this work. From now on, we furthermore abbreviate $G_p^{\mathbf{c},(2)} = G_p$ and $\Gamma_p^{(2)} = \Gamma_p$. The evolution equation (19) has a characteristic two-loop structure reminiscent of scattering diagrams in quantum field theory [21]. We note that at this stage the evolution equation is exact, such that knowledge of the four-vertex allows one to compute the exact solution for the inverse two-point functions.

For the four-vertex, we analogously obtain

$$i\partial_t \Gamma_{pqrs}^{(4)} = \Delta\omega_{pqrs}\Gamma_{pqrs}^{(4)} + V_{pqrs}(\Gamma^{(2)}) - \mathcal{M}_{pqrs}(\Gamma). \tag{20}$$

The result consists of three different contributions: The first term corresponds to the free evolution, and it is obtained by applying the four field-derivatives to the terms in the first line of Eq. (16). Corresponding terms will appear at all orders in the hierarchy of evolution equations and they lead to phase rotations with the single particle energies, $\Delta\omega_{pqrs} = \omega_p + \omega_q - \omega_r - \omega_s$, with $\omega_p = p^2/2m - \mu$. The second term is the "bare" vertex function

$$V_{pqrs} = V_{pqrs}^{\mathrm{C}} + V_{pqrs}^{\mathrm{Q}}, \tag{21}$$

which consists of a classical scattering vertex

$$V_{pqrs}^{\mathrm{C}} = -g(\Gamma_p + \Gamma_q - \Gamma_r - \Gamma_s) \equiv \quad + \text{perm.}, \tag{22}$$

as well as a quantum contribution

$$V_{pqrs}^{\mathrm{Q}} = \frac{g}{4}(\Gamma_p \Gamma_q (\Gamma_r + \Gamma_s) - \Gamma_r \Gamma_s (\Gamma_p + \Gamma_q)) \equiv \quad + \text{perm.}, \tag{23}$$

where solid lines are amputated, i.e. corresponding two-point functions are removed. Quantum scattering involves additional factors of inverse two-point functions, such that classical scattering dominates for large occupancies. We furthermore obtain higher-loop contributions $\mathcal{M}_{pqrs}(\Gamma)$, which contain interaction vertices up to sixth order $\Gamma^{(n\leq 6)}$, see Fig. 2. They originate from derivatives acting on the second line of Eq. (16), as detailed in appendix B, and here we focus on the translation invariant system. The set of diagrams couples the evolution of four-point interactions with six-point correlations as well as non-linear combinations of

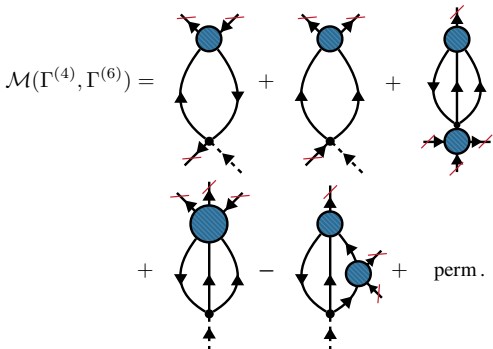

Figure 2: Loop-contributions to the evolution equation of $\Gamma^{(4)}$ involving 1PI four-
and six-vertices in a translation invariant system. Note that all external propagators
(solid lines) are amputated and permutations of legs are implied. For the underlying
analytical expressions, see B.

four-vertices to realize the complex dynamics of the Bose fields. Corresponding higher-order
evolution equations for $\Gamma^{(n\geq6)}$ follow from analogous differentiations of Eq. (16).

The vertex (21) already carries the "gain minus loss" structure characteristic for effective
ldescriptions in terms of kinetic equations. To also make contact with kinetic descriptions, we
derive the evolution equation for two-point functions from Eq. (19)

$$\partial_t G_p = -G_p(\partial_t \Gamma_p)G_p = iG_p\Big[\; \text{⟨diagram⟩} - \text{⟨diagram⟩} \;\Big]G_p = \int_{q,r,s} g\,\text{Im}(\Gamma^{(4)}_{pqrs})G_p G_q G_r G_s\,. \quad (24)$$

Here, the imaginary part originates from the structure $\sim \Gamma^{(4)}_{pqrs} - \Gamma^{(4)}_{rspq}$ in Eq. (19) and the
identity $(\Gamma^{(4)}_{pqrs})^* = \Gamma^{(4)}_{rspq}$.

## 4 Perturbative expansion

The evolution equations derived in the previous section constitute an infinite hierarchy of
equations at higher orders of the vertices. The four-vertex is coupled to the six-vertex which
will depend on the eight-vertex, etc. In practice, solving this set of equations requires us
to truncate the hierarchy, for instance at a certain order of the vertices. In this section, we
consider a perturbative expansion in powers of the coupling constant $g$.

To achieve this, we address Eq. (20) by first transforming to a rotating frame
$\tilde{\Gamma}^{(4)}_{pqrs}(t) = \exp(i\Delta\omega_{pqrs}t)\Gamma^{(4)}_{pqrs}(t)$, such that

$$i\partial_t \tilde{\Gamma}^{(4)}_{pqrs} = e^{i\Delta\omega_{pqrs}t}\left(V_{pqrs}(\Gamma^{(2)}) - \mathcal{M}_{pqrs}(\Gamma)\right). \quad (25)$$

This equation may be integrated on both sides to yield our analog of a Bethe-Salpeter equation

$$i\Gamma^{(4)}_{pqrs} = \int_{t_0}^{t} dt'\, e^{i\Delta\omega_{pqrs}(t'-t)}\left(V_{pqrs}(\Gamma^{(2)}_{t'}) - \mathcal{M}_{pqrs}(\Gamma_{t'})\right), \quad (26)$$

where we assumed the initial condition $\Gamma^{(4)}_{pqrs}(t_0) = 0$ for all momenta $p,q,r,s$. This amounts to
starting the evolution from Gaussian initial conditions which is typical for kinetic descriptions.

## 4.1 Leading order

At $\mathcal{O}(g^2)$ we focus on the bare vertex and neglect all higher-order terms $\mathcal{M}$. One can show that this represents a self-consistent power counting, as $\Gamma^{(4)}$ is sourced by bare vertex terms of order $\mathcal{O}(g)$ and hence $\mathcal{M} = \mathcal{O}(g^2)$. The contribution of loop diagrams to the evolution of two-point functions will be of order $\mathcal{O}(g^3)$. We get

$$i\Gamma^{(4)}_{pqrs}(t) = \int_{t_0}^{t} dt' e^{i\Delta\omega_{pqrs}(t'-t)} V_{pqrs}(t'). \tag{27}$$

In the following, we focus on the late-time regime where the evolution of the two-point functions is slow compared to the fast-rotating phase factor $\sim \exp(i\Delta\omega_{pqrs}(t'-t))$. Hence, we set $t_0 \to -\infty$ and the evaluation of the integral yields

$$i\Gamma^{(4)}_{pqrs}(t) = \int_{-\infty}^{t} dt' e^{i\Delta\omega_{pqrs}(t'-t)} V_{pqrs}(t') = \int_{-\infty}^{\infty} dt' \theta(t-t') e^{i\Delta\omega_{pqrs}(t'-t)} V_{pqrs}(t'). \tag{28}$$

Unless stated otherwise, integration boundaries are taken as $\pm\infty$ henceforth. Here, we employ an integral representation of the Heaviside function $\theta(x) = \int d\omega/(2\pi) i \exp(-i\omega x)/(\omega + i\epsilon)$ in the limit $\epsilon \to 0^+$. Specifically, we get

$$\Gamma^{(4)}_{pqrs}(t) = \int \frac{dt' d\omega}{2\pi} \frac{1}{\omega + i\epsilon} e^{i(\Delta\omega_{pqrs}+\omega)(t'-t)} V_{pqrs}(t'). \tag{29}$$

Using a Taylor expansion of the time-dependent bare vertex with respect to the coordinate $t$,

$$V_{pqrs}(t') = e^{(t'-t)\partial_s} V_{pqrs}(s)\big|_{s=t}, \tag{30}$$

yields the expression

$$\Gamma^{(4)}_{pqrs}(t) = \int \frac{d\omega}{2\pi} \frac{1}{\omega + i\epsilon} e^{-i\partial_\omega \partial_s} V_{pqrs}(s)\big|_{s=t} \int dt' e^{i(\Delta\omega_{pqrs}+\omega)(t'-t)}. \tag{31}$$

From the integral in the second line we obtain a Dirac $\delta$-distribution, i.e.

$$\Gamma^{(4)}_{pqrs}(t) = \int \frac{d\omega}{2\pi} \frac{1}{\omega + i\epsilon} e^{-i\partial_\omega \partial_l} V_{pqrs}(l)\big|_{l=t} \delta(\Delta\omega_{pqrs} + \omega). \tag{32}$$

This expression can be integrated by parts and rewritten in terms of a derivative with respect to $\epsilon$, which subsequently is evaluated in the limit $\epsilon \to 0$,

$$\Gamma^{(4)}_{pqrs}(t) = e^{\partial_\epsilon \partial_l} \left( \frac{V_{pqrs}(l)}{-\Delta\omega_{pqrs} + i\epsilon} \right)\bigg|_{l=t, \epsilon \to 0}. \tag{33}$$

At sufficiently late times we expect time derivatives of distribution functions to be small. This follows from the assumption that distribution functions evolve slowly at long times [20]. Specifically, higher-order terms in the expansion of the exponential function include terms as $\partial_l G_p(l) = \mathcal{O}(g^2)$, which are again higher-order in the interaction constant. At order $\mathcal{O}(g^2)$, we approximate $\exp(\partial_\epsilon \partial_l) \to 1$ and get

$$\Gamma^{(4)}_{pqrs}(t) = \frac{V_{pqrs}(t)}{-\Delta\omega_{pqrs} + i\epsilon} \equiv \text{[diagram]}. \tag{34}$$

Here, we defined a new Feynman rule representation for the solution of $\Gamma^{(4)}$ at leading order, which includes the frequency factors. Corresponding factors will appear at every subsequent coupling order as discussed next.

## 4.2 Next-to-leading order (NLO)

Analogous to the leading-order result (34), one may systematically derive higher-order contributions. To compute $\Gamma^{(4)}$ at order $\mathcal{O}(g^2)$, we consider the following loop diagrams in its evolution equation

$$\mathcal{M}_{pqrs}(\Gamma^{(4)}) = \quad + \quad + \text{perm.} + \mathcal{O}(g^3), \tag{35}$$

where the four-vertices on the right-hand side are expanded to leading-order $\mathcal{O}(g)$.

To illustrate the computation of next-to-leading order contributions to the solution of $\Gamma^{(4)}$, we focus on the first diagram in Eq. (35) next. Explicitly, we get

$$= g\Gamma_q \int_{k'} G_{k'} G_{k''} \Gamma^{(4)}_{pk'k''s}\Big|_t, \tag{36}$$

where all ingredients are evaluated at time $t$ and we introduced the shorthand notation $k'' = p - s + k'$ to abbreviate the loop momentum variable. The diagram's contribution to the solution for $\Gamma^{(4)}$ reads

$$\Gamma^{(4)}_{pqrs}(t) \supset g \int_{t_0}^{t} dt' e^{i\Delta\omega_{pqrs}(t'-t)} \Gamma_q(t') \int_{k'} G_{k'}(t') G_{k''}(t') \Gamma^{(4)}_{pk'k''s}(t'). \tag{37}$$

To compute $\Gamma^{(4)}$ at order $\mathcal{O}(g^2)$ on the left-hand side, we approximate $\Gamma^{(4)}$ at order $\mathcal{O}(g)$, as given in Eq. (27), on the right-hand side of Eq. (37). We obtain

$$\Gamma^{(4)}_{pqrs} \supset \int \frac{dt'\,d\omega\,dt''\,d\omega'}{(2\pi)^2} \Gamma_q(t') \int_{k'} G_{k'}(t') G_{k''}(t') \frac{e^{i(\Delta\omega_{pqrs}+\omega)(t'-t)}}{\omega + i\epsilon} \frac{e^{i(\Delta\omega_{pk'k''s}+\omega')(t''-t')}}{\omega' + i\epsilon'} V_{pk'k''s}(t''), \tag{38}$$

where we again have set $t_0 \to -\infty$ and used the integral representation of the heaviside function. Next, we rewrite the expression in analogy to the steps performed in Eqs. (30)-(33) to arrive for the right-hand side of Eq. (38) at

$$e^{(\partial_{\epsilon'}-\partial_\epsilon)\partial_{l'}} e^{\partial_\epsilon \partial_l} \left( \frac{\Gamma_q(l) \int_{k'} G_{k'}(l) G_{k''}(l)}{\Delta\omega_{pqrs} - i\epsilon} \frac{V_{pk'k''s}(l')}{\Delta\omega_{pk'k''s} - i\epsilon'} \right)\Bigg|_{l=l'=t,\epsilon'=\epsilon\to 0}. \tag{39}$$

At order $\mathcal{O}(g^2)$, we again approximate the exponential operators by unity. Similar to Eq. (34), we identify this result after time integration with a diagrammatic expression

$$\Gamma^{(4)}_{pqrs} \supset \frac{\Gamma_q \int_{k'} G_{k'} G_{k''}}{\Delta\omega_{pqrs} - i\epsilon} \frac{V_{pk'k''s}}{\Delta\omega_{pk'k''s} - i\epsilon} \equiv \quad, \tag{40}$$

where all quantities are evaluated at time $t$. It is important to keep track of the frequency factors. In the present case, there is one factor $1/(-\Delta\omega_{pk'k''s} + i\epsilon)$ which comes with the vertex "$V$" and another factor $1/(-\Delta\omega_{pqrs} + i\epsilon)$ carrying the external momentum labels of the

left-hand side's vertex $\Gamma^{(4)}_{pqrs}$. At order $\mathcal{O}(g^2)$ we then write the solution for the time-dependent four-vertex diagrammatically as

$$\Gamma^{(4)}_{pqrs} = \;\; + \;\; + \;\; + \text{perm.} \tag{41}$$

In general, frequency factors enter the calculation for each "insertion" of $\Gamma^{(4)}$ as demonstrated for the present order in Eqs. (37) and (38).

### 4.3 Boltzmann equation

To derive the late-time evolution equation for the two-point functions at leading order ($\partial_t G_p = \mathcal{O}(g^2)$), we consider the imaginary part of the corresponding solution of the four-vertex

$$\text{Im}\left(\Gamma^{(4)}_{pqrs}(t)\right) = -\pi\delta(\Delta\omega_{pqrs})V_{pqrs}(t). \tag{42}$$

Here, the imaginary part is a crucial ingredient to obtain the energy conservation of the particles which stream freely in-between collisions. Plugging this result into Eq. (24), one finds

$$\partial_t f_p = \frac{g^2}{2}\int_{q,r,s} (2\pi)^3\delta(p+q-r-s)(2\pi)\delta(\Delta\omega_{pqrs})$$
$$\times \left((f_p+1)(f_q+1)f_r f_s - f_p f_q(f_r+1)(f_s+1)\right), \tag{43}$$

which is the well-known Boltzmann equation for weakly correlated non-relativistic systems. Here, we defined a distribution function $G_p = f_p + 1/2$ as in Eq. (7) corresponding to $f_p = \langle\hat{\psi}^\dagger_p\hat{\psi}_p\rangle$, with $\hat{\psi}_p$ being the Fourier transformed field operator. Using this, one finds

$$V_{pqrs}G_p G_q G_r G_s = g[f_p f_q(f_r+1)(f_s+1) - (f_p+1)(f_q+1)f_r f_s], \tag{44}$$

which yields the result (43). It contains a characteristic "gain minus loss" structure and has a momentum independent scattering rate $g^2/2$, such that we get the matrix element $|T_{pqrs}|^2 = g^2/2(2\pi)^3\delta(p+q-r-s)(2\pi)\delta(\Delta\omega_{pqrs})$. The equation describes a dilute medium with occupancy $f_p \sim \mathcal{O}(1)$ for weak coupling at sufficiently late times, where leading-order perturbation theory is expected to be valid. In the following section we derive the corresponding scattering rate for a non-perturbative setting which allows to also access the regime of over-occupied Bose fields.

## 5 Non-perturbative large-$N$ expansion

While the previous section dealt with a perturbative expansion of equal-time vertices, we consider a non-perturbative expansion for large numbers of field components next [26, 27]. Starting from the corresponding flow equation, the expansion will allow us to sum an infinite number of scattering interactions as shown below. This yields important corrections to the equal-time effective vertices, which can drastically alter the dynamics of Bose fields in strongly correlated regimes.

The large-$N$ counting scheme is detailed in appendix C. We use in the following that for suitable initial conditions, for instance Gaussian states, equal-time vertices obey [28]

$$\Gamma^{(n)} = \mathcal{O}\left(\frac{1}{N^{\frac{n-2}{2}}}\right), \quad n > 2. \tag{45}$$

Specifically, for Gaussian initial states, where $\Gamma^{(n>2)}(t_0) = 0$, the evolution of $\Gamma^{(4)}$ is sourced by the bare vertex (see Eq. (1)) at order $\mathcal{O}(1/N)$. Vertices $\Gamma^{(n>4)}$ subsequently build up at corresponding higher orders through combinations of bare vertices and $\Gamma^{(4)}$ according to Eq. (45). Then, loop diagrams as displayed in Fig. 2 also contribute to the evolution of $\Gamma^{(4)}$ at order $\mathcal{O}(1/N)$ at most, where every vertex comes with a factor of $1/N$ and factors of $N$ originate from summation over field components in closed loops. To determine the contribution of equal-time vertices to the evolution equation at order $1/N$, we focus on the case, with external field indices $i_1 = i_4$ and $i_2 = i_3$.

In the following, to keep the notation in the main text simple, we use the U($N$) symmetry to diagonalize the two-point function in field space. Subsequently, we may explicitly sum over the field components, and we will omit the field index $i$ in our notation. The bare vertex is given by

$$V_{pqrs} = -\frac{g}{2N}(\Gamma_p^{(2)} + \Gamma_q^{(2)} - \Gamma_r^{(2)} - \Gamma_s^{(2)}) + \frac{g}{8N}\left(\Gamma_p^{(2)}\Gamma_q^{(2)}(\Gamma_r^{(2)} + \Gamma_s^{(2)}) - \Gamma_r^{(2)}\Gamma_s^{(2)}(\Gamma_p^{(2)} + \Gamma_q^{(2)})\right), \tag{46}$$

and using the expansion to order $\mathcal{O}(1/N)$, the evolution equation for $\Gamma^{(4)}$ involves no more than propagators and four-vertices, i.e.

$$\mathcal{M}_{pqrs}(\Gamma^{(4)}) = \text{[diagram]} + \text{perm.} + \mathcal{O}\left(\frac{1}{N^2}\right), \tag{47}$$

where

$$\text{[diagram]} = \frac{g}{2}\Gamma_q \int_{k'} G_{k'}G_{k''}\Gamma_{pk'k''s}^{(4)}\Big|_t = \mathcal{O}\left(\frac{1}{N}\right). \tag{48}$$

The shown diagram involves two factors of $1/N$ for the bare vertex and for $\Gamma^{(4)}$, as well as a factor of $N$ representing the different field components which "run" in the loop (cf. appendix C). The corresponding evolution equation thus evolves $\Gamma^{(4)}$ again at order $1/N$, and it can formally be integrated to yield

$$i\Gamma_{pqrs}^{(4)}(t) = \int_{t_0}^{t} dt' e^{i\Delta\omega_{pqrs}(t'-t)}\left(V_{pqrs}(t') - \mathcal{M}_{pqrs}(t')\right), \tag{49}$$

which by employing analogous approximations as in the previous section becomes

$$\Gamma_{pqrs}^{(4)}(t) = \frac{-1}{\Delta\omega_{pqrs} - i\epsilon}\left(V_{pqrs}(t) + \left[\frac{g}{2}(\Gamma_q - \Gamma_r)\int_{k'} G_{k'}G_{k''}\Gamma_{pk'k''s}^{(4)}(t) + \{p, s \leftrightarrow q, r\}\right]\right), \tag{50}$$

where we sum over a second term with permuted external legs. For $i_1 = i_4$ and $i_2 = i_3$ at order $1/N$, only combined permutations of $p, s$ with $q, r$ appear. At this stage, Eq. (50) is

analogous to Eq. (37), but we keep $\Gamma^{(4)}$ consistently at order $1/N$ here. Eq. (50) can be solved by iteration in terms of an infinite set of loop diagrams. In the following we first illustrate the iterative computation to two-loop order, while we subsequently calculate the full evolution of distribution functions at order $1/N$.

The vertex is diagrammatically given by

$$
\Gamma^{(4)}_{pqrs} = \quad\vcenter{\hbox{\includegraphics{}}}\quad + \quad\vcenter{\hbox{\includegraphics{}}}\quad + \text{perm.} + \sum_{n=2}^{\infty} n\text{-loop} \tag{51}
$$

$$
= \frac{-1}{\Delta\omega_{pqrs} - i\epsilon}\left(V_{pqrs} - \left[\frac{g}{2}(\Gamma_q - \Gamma_r)\int_{k'} G_{k'}G_{k''}\frac{V_{pk'k''s}}{\Delta\omega_{pk'k''s} - i\epsilon} + \{p, s \leftrightarrow q, r\}\right]\right)
$$

$$
+ \text{higher-orders},
$$

where frequency factors are assigned analogous to Eq. (40). The corresponding two-loop expression is given by summing the following diagrams

$$
2\text{-loop} = \quad\vcenter{\hbox{\includegraphics{}}}\quad + \quad\vcenter{\hbox{\includegraphics{}}}\quad + \quad\vcenter{\hbox{\includegraphics{}}}\quad + \text{perm.}, \tag{52}
$$

where summation over permutations of external lines is implied. The first diagram corresponds to the equation

$$
\vcenter{\hbox{\includegraphics{}}} \equiv \left(\frac{g}{2}\right)^2\int_{k',q'}\frac{-\Gamma_r}{\Delta\omega_{pqrs} - i\epsilon}\frac{G_{k'}}{\Delta\omega_{pk'k''s} - i\epsilon}\frac{G_{q'}G_{q''}V_{pq'q''s}}{\Delta\omega_{pq'q''s} - i\epsilon}, \tag{53}
$$

where the two-point function at momentum $k''$ is amputated by the inverse propagator represented by the dashed line. The first frequency factor $1/(-\Delta\omega_{pqrs} + i\epsilon)$ originates from the external lines, the second factor carries the momentum labels of lines connecting to the upper loop, i.e. $p$, $k'$, $k''$, and $s$. The last insertion is given by the bare vertex, which comes with a frequency factor carrying the same momentum labels, $1/(-\Delta\omega_{pq'q''s} + i\epsilon)$. Analogously, the last diagram is obtained as

$$
\vcenter{\hbox{\includegraphics{}}} \equiv \left(\frac{g}{2}\right)^2\int_{q',k'}\frac{-\Gamma_r\Gamma_s}{\Delta\omega_{pqrs} - i\epsilon}\frac{G_{k'}G_{k''}}{\Delta\omega_{pk'k''s} - i\epsilon}\frac{G_{q'}G_{q''}V_{k'q''q'k''}}{\Delta\omega_{k'q''q'k''} - i\epsilon}. \tag{54}
$$

Again, the result is augmented with frequency factors for each iteration step, which carry ingoing and out-going momentum labels according to the momenta of the internal four-vertex which is iterated. Here, the last insertion of the bare vertex is internal, i.e. the corresponding frequency factor carries internal momenta only. In general, one needs to keep track of the "history" of insertions for the correct assignment of labels.

At this point, we distinguish different cases originating from the various possibilities of forming diagrams at order $1/N$. In the following, we sort contributions according to the number of external inverse two-point functions attached to the diagram. To this end, we will name the set of loop diagrams leading to an odd number of external inverse propagators $\Gamma^A$, while the loop diagrams with an even number are represented by $\Gamma^B$. Thus, $\Gamma^A$ represents important corrections to the bare classical and quantum vertices, while $\Gamma^B$ yields a new type of vertex which is not present in the perturbative theory. The summation of both contributions will yield a Boltzmann equation for a strongly-correlated Bose system with momentum- and medium-dependent scattering rate. We note that the series of diagrams at order $1/N$ is reminiscent of the diagrams employed in Ref. [29], where (unequal-time) propagators are similarly sorted by their quantum (dashed) and classical (solid) external lines.

## 5.1 Vertex $\Gamma^A$

At first, we consider the terms with an odd number of external propagators. To distinguish the diagrams with respect to their configuration of external legs we introduce the vertices

$$\frac{V^C_{pqrs}}{-\Delta\omega_{pqrs}+i\epsilon} \equiv \quad + \text{ perm.} \tag{55}$$

and

$$\frac{V^Q_{pqrs}}{-\Delta\omega_{pqrs}+i\epsilon} \equiv \quad + \text{ perm.}, \tag{56}$$

see also the definition of Eq. (34). The expression $\Gamma^A$ is given by the sum of the bare vertices (55) and (56) with all $n$-loop diagrams $\Gamma^{A,n}$ involving an odd number of external $\Gamma^{(2)}$, i.e.

$$\Gamma^A_{pqrs}(t) = \quad + \quad + \text{ perm.} + \sum_{n=1}^{\infty} \Gamma^{A,n}_{pqrs}(t)$$

$$\equiv \quad + \quad + \text{ perm.}, \tag{57}$$

where the symbol "A" comprises all diagrams with according configuration of external legs. For a particular configuration, Eq. (50) reads diagrammatically

$$= \quad + \quad + \quad . \tag{58}$$

Iterating this equation for example to one-loop order, we obtain

$$+ \quad = -\frac{1}{N}\frac{\Gamma_r}{\Delta\omega_{pqrs}-i\epsilon}\Pi_{ps}, \tag{59}$$

including the usual frequency factors. Here, we also defined a one-loop self-energy function as

$$\Pi_{ps}(t) = \frac{g}{2}\int_{k'}\frac{G_{k'}(t)-G_{k''}(t)}{\Delta\omega_{pk''s}-i\epsilon}, \tag{60}$$

where we sum over both possibilities of amputating an internal loop propagator. Using the series $\sum_{n\geq 0}(-\Pi_{ps})^n = 1/(1+\Pi_{ps})$, the sum over all loop orders reads

$$\Gamma^A_{pqrs} = -\frac{V_{p\underline{q}rs}}{\Delta\omega_{pqrs} - i\epsilon} \times \frac{1}{1+\Pi_{ps}} + \{p,s \leftrightarrow q,r\}, \tag{61}$$

where we introduced the notation

$$V_{p\underline{q}rs}/g = -\frac{1}{2N}(\Gamma_q - \Gamma_r)\left(1 - \frac{1}{4}\Gamma_p\Gamma_s\right), \tag{62}$$

$$V_{\underline{p}qr\underline{s}}/g = -\frac{1}{2N}(\Gamma_p - \Gamma_s)\left(1 - \frac{1}{4}\Gamma_q\Gamma_r\right), \tag{63}$$

i.e. $V_{pqrs} = V_{p\underline{q}rs} + V_{\underline{p}qr\underline{s}}$.

The vertex $\Gamma^A$ has a similar structure as the bare vertex defined in Eq. (34), including a correction arising from the non-perturbative resummation of the infinite series of diagrams presented in this section. Indeed, the bare vertex $V$ emerges from Eq. (61) in the perturbative expansion at leading order, where $\Pi = \mathcal{O}(g)$ and hence $1/(1+\Pi) \to 1 + \mathcal{O}(g)$.

The relevant contribution to the evolution equation (24) is the imaginary part of the four-vertices. Here, we get

$$\text{Im}\left(\Gamma^A_{pqrs}\right) = -\pi\delta(\Delta\omega_{pqrs})\text{Re}\left(\frac{V_{p\underline{q}rs}}{1+\Pi_{ps}}\right) - \mathcal{P}\left[\frac{1}{\Delta\omega_{pqrs} - i\epsilon}\right]\text{Im}\left(\frac{V_{p\underline{q}rs}}{1+\Pi_{ps}}\right) + \{p,s \leftrightarrow q,r\}, \tag{64}$$

where $\mathcal{P}$ denotes the Cauchy principal value. Using the identities $\text{Im}(1/(1+\Pi)) = -\text{Im}(\Pi)/|1+\Pi|^2$ and $\text{Re}(1/(1+\Pi)) = (1+\text{Re}(\Pi))/|1+\Pi|^2$, and the definition of an effective coupling and vertex

$$g^{\text{eff}}_{ps} = \frac{g}{|1+\Pi_{ps}|^2}, \qquad V^{\text{eff}}_{pqrs} = \frac{V_{pqrs}}{|1+\Pi_{ps}|^2}, \tag{65}$$

we find

$$\text{Im}\left(\Gamma^A_{pqrs}\right) = -\pi\delta(\Delta\omega_{pqrs})V^{\text{eff}}_{pqrs}\left(1 + \mathcal{P}(\Pi_{ps})\right) + \mathcal{P}\left[\frac{1}{\Delta\omega_{pqrs} - i\epsilon}\right]\left[V^{\text{eff}}_{p\underline{q}rs}\text{Im}\left(\Pi_{ps}\right) + \{p,s \leftrightarrow q,r\}\right]. \tag{66}$$

In the first line, we used that $\text{Re}(\Pi_{qr}) = \text{Re}(\Pi_{ps})$ under the conditions of energy and momentum conservation, represented by $\delta(\Delta\omega_{pqrs})\delta(p+q-r-s)$, see appendix F. We thus find an on-shell contribution $\sim \delta(\Delta\omega_{pqrs})$ as well as off-shell terms involving the principle value $\mathcal{P}\left(1/(\Delta\omega_{pqrs} - i\epsilon)\right)$. All terms include the effective coupling $g^{\text{eff}}$, which leads to a suppression of the effective interaction of modes if the gas is highly occupied towards lower momenta [30]. In this regime, the denominator is dominated by the large distribution function in the one-loop self-energy $\Pi$ [24]. Our findings are in qualitative agreement with results obtained from Ref. [2], where such an infrared suppression was observed experimentally.

## 5.2 Vertex $\Gamma^B$

Next, we consider the diagrams with an even number of external inverse propagators and (amputated) propagators. We may similarly sort terms by their number of loops

$$\Gamma^B_{pqrs}(t) = \sum_{n=1}^{\infty}\Gamma^{B,n}_{pqrs}(t) \equiv \raisebox{-1.5em}{\includegraphics[height=3em]{diagram}} + \text{perm.}, \tag{67}$$

however, no bare vertices appear in this case, and we define the diagram "B" according to the configuration of the two external $\Gamma^{(2)}$. In terms of bare vertices, we get the series

$$
\text{(diagram)} = \text{(diagram)} + \text{(diagram)} + \text{(diagram)} + \text{(diagram)} + \{p, s \leftrightarrow q, r\} + \text{higher-order loops}. \tag{68}
$$

For any loop order, this equation may equivalently be written in the form

$$
\text{(diagram)} = \text{(diagram)} + \text{(diagram)} + \text{(diagram)} + \text{(diagram)}
$$
$$
+ \{p, s \leftrightarrow q, r\} + \text{higher-order loops}. \tag{69}
$$

Explicitly, for a general configuration of external legs, we obtain the integral equation

$$
\Gamma^B_{pqrs} = \frac{1}{N}\left(\frac{g}{2}\right)^2 \frac{(\Gamma_p - \Gamma_s)(\Gamma_q - \Gamma_r)}{-\Delta\omega_{pqrs} + i\epsilon} \int_{k'} \left( -\frac{G_{k'}G_{k''} - \frac{1}{4}}{\Delta\omega_{k'psk''} - i\epsilon} \frac{1}{1 + \Pi_{k'k''}} - \frac{G_{k'}G_{k''} - \frac{1}{4}}{\Delta\omega_{k''qrk'} - i\epsilon} \frac{1}{1 + \Pi_{k''k'}} \right)
$$
$$
+ \frac{g}{2}\frac{\Gamma_r - \Gamma_q}{-\Delta\omega_{pqrs} + i\epsilon} \int_{k'} G_{k'}G_{k''}\Gamma^B_{pk'k''s} + \frac{g}{2}\frac{\Gamma_s - \Gamma_p}{-\Delta\omega_{pqrs} + i\epsilon} \int_{k'} G_{k'}G_{k''}\Gamma^B_{k''qrk'}, \tag{70}
$$

where we used the solution for $\Gamma^A$, Eq. (61). In the following, we focus on the imaginary part of Eq. (70), as $\text{Im}(\Gamma^B)$ is the relevant quantity to determine the evolution of two-point functions in the Bose system. In appendix D, we demonstrate that this equation is solved by the ansatz

$$
\text{Im}(\Gamma^B_{pqrs}) = \mathcal{P}\left[ \frac{(\Gamma_p - \Gamma_s)(\Gamma_q - \Gamma_r)}{-\Delta\omega_{pqrs} + i\epsilon} \right] \left( \frac{g\, g^{\text{eff}}_{ps}}{4N} \int_{k'} \pi\delta(\Delta\omega_{pk'k''s})\left(G_{k'}G_{k''} - \frac{1}{4}\right) + \{p, s \leftrightarrow q, r\} \right). \tag{71}
$$

Again, we find that each term contains the medium-augmented non-perturbative interaction vertex $g^{\text{eff}}$ through the resummation of diagrams.

## 5.3 Non-perturbative Boltzmann equation

In this section we assemble the results for the resummed effective vertices to derive the evolution equation for two-point functions. Starting from Eq. (24), one gets

$$
\partial_t G_p = \int_{q,r,s} g\,\text{Im}(\Gamma^{(4)}_{pqrs})G_p G_q G_r G_s = \int_{q,r,s} g\,\text{Im}(\Gamma^A_{pqrs} + \Gamma^B_{pqrs})G_p G_q G_r G_s, \tag{72}
$$

where momentum conservation, represented by $\delta(p + q - r - s)$, is implied. The full vertex solution is the sum over all configurations of external legs and hence we add $\Gamma^A$ and $\Gamma^B$. Plugging in Eqs. (66) and (71) we obtain by direct computation (appendix E)

$$
\partial_t G_p = -\int_{q,r,s} \pi\delta(\Delta\omega_{pqrs}) V^{\text{eff}}_{pqrs} G_p G_q G_r G_s, \tag{73}
$$

which is equivalent to the previous perturbative Boltzmann equation except for the replacement $V \to V^{\text{eff}}$, i.e.

$$\partial_t f_p = \int_{q,r,s} \frac{g g_{ps}^{\text{eff}}}{4N} (2\pi)^3 \delta(p+q-r-s)(2\pi)\delta(\Delta\omega_{pqrs})\big((f_p+1)(f_q+1)f_r f_s - f_p f_q (f_r+1)(f_s+1)\big). \quad (74)$$

Accordingly, the matrix element of the kinetic equation is given by $|T_{pqrs}|^2 = g g_{ps}^{\text{eff}}/(4N)(2\pi)^3 \times \delta(p+q-r-s)(2\pi)\delta(\Delta\omega_{pqrs})$, which receives the momentum-dependent correction $1/|1+\Pi_{ps}|^2$ compared to the perturbative case. This momentum dependence dominates the non-perturbative evolution with large occupations where $\Pi \gg 1$, with drastic consequences for dynamical phenomena such as turbulence and far-from-equilibrium universality [12, 30, 31].

# 6 Measurement protocol

The above coupling and large-$N$ expansion results establish a direct link between equal-time correlations and standard observables for effective kinetic theories and corresponding hydrodynamic descriptions. However, the quantum evolution equations we derive in section 3 from the equal-time effective action are exact and not limited to kinetic theory approximations. For instance, the exact time evolution equation (19) relates the time derivative of $\Gamma^{(2)}$ – encoding distribution information – to the effective interaction $\Gamma^{(4)}$ and convolutions with the distributions. It would be a tremendous progress for quantum many-body physics to be able to extract the exact quantum evolution equation for strongly correlated systems from quantum simulation measurements of $\Gamma^{(2)}$ and $\Gamma^{(4)}$ for relevant times. This would provide important insights into the long-standing problem of finding suitable approximations of the time evolution equations also for strongly coupled systems and their range of validity.

In this section we devise an efficient scheme to measure equal-time effective vertices, in particular $\Gamma^{(2)}$ and $\Gamma^{(4)}$, in cold-atom quantum simulators. This discussion extends the procedures of Refs. [1, 2] to the underlying Bose fields appearing in the defining Hamiltonian. Often, such experiments are limited to extracting equal-time density correlations. A common strategy is to let the system evolve to time $t$ and illuminate with light to obtain a snapshot at this instant of time. Subsequently, density correlations are extracted by averaging over many repetitions of this procedure.

To relate the experiment with theory it is especially beneficial to express effective descriptions of the system in such equal-time quantities. We hence provide a protocol to extract the relevant two- and four-point correlation functions via density measurements. 1PI equal-time vertices are extracted according to Eq. (12), i.e. by "amputating" the external legs. This is most efficiently performed in Fourier space, where "amputation" refers to dividing out the corresponding two-point functions. While we illustrate our scheme for the case of a single-component gas, it is more general and can similarly be applied to multi-component systems.

The desired quantities are the symmetrized two- and four-point correlation functions of the fields, i.e. $\langle\{\hat{\psi}_x^\dagger, \hat{\psi}_y\}\rangle$ and $\langle\hat{\psi}_x^\dagger \hat{\psi}_v^\dagger \hat{\psi}_y \hat{\psi}_w\rangle_{\text{sym}}$. While we specifiy the symmetrically ordered correlation functions here, all other operator orderings are equivalent and they are related through the equal-time commutation relations. For simplicity, we focus here on the observables

$$\mathcal{O}_1 = \langle\hat{\mathcal{O}}_1\rangle = \langle\hat{\psi}_x^\dagger \hat{\psi}_y\rangle, \quad (75)$$

$$\mathcal{O}_2 = \langle\hat{\mathcal{O}}_2\rangle = \langle\hat{\psi}_x^\dagger \hat{\psi}_v^\dagger \hat{\psi}_y \hat{\psi}_w\rangle. \quad (76)$$

The central idea is to couple the atoms at the respective positions to ancilla degrees of freedom to create effective three-state systems, for instance in a $\Lambda$-type configuration, see Fig. 3. Raman

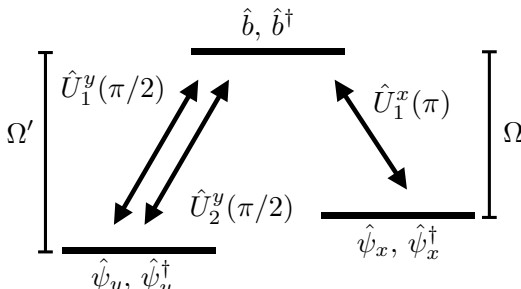

Figure 3: Schematics of the measurement scheme. We construct an effective Λ-type level scheme and propose to introduce rotations between the states with Raman transitions or microwave coupling with strengths $\Omega, \Omega'$.

beams or microwave pulses can transfer population among the three states or manipulate their relative phases. Achieving this requires addressing the bosonic fields position-selectively, e.g. by locally adjusting the chemical potential $\mu$.

We start by transferring the population from position $x$ to the ancilla. Afterwards we couple position $y$ to the ancilla, therefore effectively coupling positions $x$ and $y$. Density measurements of ancilla and the spatial mode at $y$ will then give rise to $\langle \hat{\mathcal{O}}_1 \rangle$. Here we use that these manipulations can be performed on much shorter time-scales compared to the system evolution, such that the information about the quantum state is effectively frozen during the measurement procedure. We outline this procedure in detail next.

We describe the ancilla degree of freedom "b" with bosonic operators $\hat{b}, \hat{b}^\dagger$. The microwave interaction of the ancilla with a bosonic mode $\hat{\psi}_x, \hat{\psi}_x^\dagger$ is described by the two unitary operators

$$\hat{U}_1^x(\varphi) = e^{i\varphi(\hat{b}^\dagger \hat{\psi}_x + \hat{\psi}_x^\dagger \hat{b})}, \tag{77}$$

$$\hat{U}_2^x(\varphi) = e^{\varphi(\hat{b}^\dagger \hat{\psi}_x - \hat{\psi}_x^\dagger \hat{b})}. \tag{78}$$

In a Schwinger boson representation these operators may locally be interpreted as rotations of the collective spin on a Bloch sphere. Using bosonic commutation relations, we obtain the transformations of the operators

$$\hat{\psi}_x \rightarrow \hat{U}_1^x(\varphi)\hat{\psi}_x(\hat{U}_1^x(\varphi))^\dagger = \cos(\varphi)\hat{\psi}_x + i\sin(\varphi)\hat{b}, \tag{79}$$

$$\hat{\psi}_x \rightarrow \hat{U}_2^x(\varphi)\hat{\psi}_x(\hat{U}_2^x(\varphi))^\dagger = \cos(\varphi)\hat{\psi}_x + \sin(\varphi)\hat{b}. \tag{80}$$

We first couple the ancilla to the atoms at position $x$ with $\hat{U}_1^x(\pi)$. Secondly, we couple position $y$ to the ancilla with $\hat{U}_1^y(\pi/2)$. A subsequent density measurement yields

$$\langle \hat{b}^\dagger \hat{b} \rangle_1 = \langle \psi(t)| \hat{U}_1^y(\pi/2)\hat{U}_1^x(\pi)\hat{b}^\dagger \hat{b}\, \hat{U}_1^{x,\dagger}(\pi)\hat{U}_1^{y,\dagger}(\pi/2) |\psi(t)\rangle$$

$$= \frac{1}{2}(\langle \hat{\psi}_x^\dagger \hat{\psi}_x \rangle + \langle \hat{\psi}_y^\dagger \hat{\psi}_y \rangle) - \text{Im}(\mathcal{O}_1), \tag{81}$$

where $|\psi(t)\rangle$ is the time-dependent Schrödinger quantum state, and $\langle \hat{\psi}_x^\dagger \hat{\psi}_x \rangle$ is the local mean number density at time $t$, which can be accessed in a separate measurement. Measuring the density at position $y$ after the rotation $\hat{U}_1$ similarly yields

$$\langle \hat{\psi}_y^\dagger \hat{\psi}_y \rangle_1 = \langle \psi(t)| \hat{U}_1^y(\pi/2)\hat{\psi}_y^\dagger \hat{\psi}_y\, \hat{U}_1^{y,\dagger}(\pi/2) |\psi(t)\rangle$$

$$= \frac{1}{2}(\langle \hat{\psi}_x^\dagger \hat{\psi}_x \rangle + \langle \hat{\psi}_y^\dagger \hat{\psi}_y \rangle) + \text{Im}(\mathcal{O}_1), \tag{82}$$

such that we obtain $\text{Im}(\mathcal{O}_1)$ by subtracting the two measurements. In order to measure the corresponding real part in separate realizations of the experiment, we perform the same series of unitary operations with $\hat{U}_2$ replacing the second operator. We get

$$
\begin{aligned}
\langle \hat{b}^\dagger \hat{b} \rangle_2 &= \langle \psi(t)| \, \hat{U}_2^y(\pi/2)\hat{U}_1^x(\pi)\hat{b}^\dagger \hat{b} \, \hat{U}_1^{x,\dagger}(\pi)\hat{U}_2^{y,\dagger}(\pi/2) \, |\psi(t)\rangle \\
&= \frac{1}{2}(\langle \hat{\psi}_x^\dagger \hat{\psi}_x \rangle + \langle \hat{\psi}_y^\dagger \hat{\psi}_y \rangle) + \text{Re}(\mathcal{O}_1) \,,
\end{aligned}
\tag{83}
$$

and for position $y$ one finds

$$
\begin{aligned}
\langle \hat{\psi}_y^\dagger \hat{\psi}_y \rangle_2 &= \langle \psi(t)| \, \hat{U}_2^y(\pi/2)\hat{\psi}_y^\dagger \hat{\psi}_y \, \hat{U}_2^{y,\dagger}(\pi/2) \, |\psi(t)\rangle \\
&= \frac{1}{2}(\langle \hat{\psi}_x^\dagger \hat{\psi}_x \rangle + \langle \hat{\psi}_y^\dagger \hat{\psi}_y \rangle) - \text{Re}(\mathcal{O}_1) \,.
\end{aligned}
\tag{84}
$$

The difference yields the real part $\text{Re}(\mathcal{O}_1)$, which subsequently allows to reconstruct the observable $\mathcal{O}_1$ by combining real and imaginary part. In the Schwinger boson representation real and imaginary parts of $\mathcal{O}_1$ correspond to spin projections in $x$ and $y$ direction, respectively. Our scheme thus effectively employs appropriate spin rotations to rotate the information to the $z$ component which can be accessed with density measurements.

Similarly, one can access the observable $\mathcal{O}_2$ by adding a second ancilla mode "d" with bosonic operators $\hat{d}, \hat{d}^\dagger$ and measuring density correlations between both ancillas. Here, we focus on the situation where all four positions $x, v, y, w$ are different, assuming those cases with equal positions to become irrelevant in the thermodynamic limit of a large-scale quantum system described by quantum field theory. We assign ancilla b to positions $x, y$ and ancilla d to positions $v, w$ and subsequently perform the same operations on both sets of modes individually. Appropriate density correlation measurements of both ancillas give the four combinations

$$
\left\langle \left\{ \text{Re}\big(\hat{\mathcal{O}}_1(x,y)\big), \text{Re}\big(\hat{\mathcal{O}}_1(v,w)\big) \right\} \right\rangle,
\tag{85}
$$

$$
\left\langle \left\{ \text{Re}\big(\hat{\mathcal{O}}_1(x,y)\big), \text{Im}\big(\hat{\mathcal{O}}_1(v,w)\big) \right\} \right\rangle,
\tag{86}
$$

$$
\left\langle \left\{ \text{Im}\big(\hat{\mathcal{O}}_1(x,y)\big), \text{Re}\big(\hat{\mathcal{O}}_1(v,w)\big) \right\} \right\rangle,
\tag{87}
$$

$$
\left\langle \left\{ \text{Im}\big(\hat{\mathcal{O}}_1(x,y)\big), \text{Im}\big(\hat{\mathcal{O}}_1(v,w)\big) \right\} \right\rangle,
\tag{88}
$$

where $\{\cdot, \cdot\}$ is the anti-commutator. These contributions can be combined with the equal-time commutation relations and two-point functions $\mathcal{O}_1$ to compute the observable $\mathcal{O}_2$.

## 7 Conclusion

In our work we outlined an equal-time approach to the dynamics of quantum fields out of equilibrium. Starting from the equal-time quantum effective action, we derived effective kinetic equations in two regimes: First we considered a dilute, perturbative system governed by two-to-two scattering. Secondly, we extended our analysis to the case of non-perturbatively large occupancies of the gas, which results in important vertex corrections to the scattering rates.

Our results open up new avenues in non-equilibrium quantum field theory. While our calculations are performed for a non-relativistic Bose gas, the approach is general and can also be applied to systems with fermions, relativistic field theories, and in particular gauge theories where the time-local formulation can provide important advantages in finding approximations consistent with local gauge symmetries. Moreover, the textbook ("unequal-time") approach

to non-equilibrium quantum field theory employing a closed-time path yields ab initio evolution equations that are non-local in time, such that late times are difficult to reach and the derivation of efficient time-local descriptions require additional approximations.

Most importantly, our approach matches experimental capabilities of quantum simulators such as employing ultra-cold quantum gases. These platforms offer the unique opportunity to extract the irreducible correlations directly from experiments in the many-body regime described by quantum fields. Our results demonstrate that the extraction of lower equal-time correlations, such as the two- and four-point functions, involve already all the ingredients to obtain effective kinetic descriptions from first principles. Strikingly, the approach also offers the perspective of determining the exact evolution equations from quantum simulations. This can provide essential insights into the long-standing problem of finding non-perturbative approximations for strongly coupled systems.

## Acknowledgments

We thank Sebastian Erne, Gregor Fauth, Michael Heinrich, Markus Oberthaler, Maximilian Prüfer, and Jörg Schmiedmayer for fruitful discussions and collaboration on related work. This work is funded by the DFG (German Research Foundation) under Project-ID 27381115 – SFB 1225 ISOQUANT, and under Germany's Excellence Strategy EXC2181/1-390900948 – the Heidelberg STRUCTURES Excellence Cluster. This work was supported by the Simons Collaboration on UltraQuantum Matter, which is a grant from the Simons Foundation (651440, P.Z.).

## A  Diagrammatic examples

In this appendix, we present further details on the calculations given in the main text.

To illustrate the diagrammatic rules Eqs. (13)-(15), we give the full expression for the exemplary diagrams of Fig. 1 for the example of $N = 1$. The first diagram represents the connected four-point function (Eq. (12)), which is obtained as

$$
\begin{aligned}
G^{\mathbf{c},(4)}_{x_1 x_2 x_3 x_4} &= \frac{\delta}{\delta J^*_{x_1}} \frac{\delta}{\delta J^*_{x_2}} \frac{\delta}{\delta J_{x_3}} \frac{\delta}{\delta J_{x_4}} W_t[J^{(*)}] \Big|_{J,J^*=0} \\
&= -\int_{\mathbf{y}} \frac{\delta \psi_{y_1}}{\delta J^*_{x_1}} \frac{\delta \psi_{y_2}}{\delta J^*_{x_2}} \frac{\delta \psi_{y_3}}{\delta J_{x_3}} \frac{\delta \psi_{y_4}}{\delta J_{x_4}} \Gamma^{(4)}_{y_1 y_2 y_3 y_4} \\
&= -\int_{\mathbf{y}} \frac{\delta W_t}{\delta J^*_{x_1} \delta J_{y_1}} \frac{\delta W_t}{\delta J^*_{x_2} \delta J_{y_2}} \frac{\delta W_t}{\delta J^*_{y_3} \delta J_{x_3}} \frac{\delta W_t}{\delta J^*_{y_4} \delta J_{x_4}} \Gamma^{(4)}_{y_1 y_2 y_3 y_4} \\
&= -\int_{\mathbf{y}} G^{\mathbf{c},(2)}_{x_1 y_1} G^{\mathbf{c},(2)}_{x_2 y_2} G^{\mathbf{c},(2)}_{y_3 x_3} G^{\mathbf{c},(2)}_{y_4 x_4} \Gamma^{(4)}_{y_1 y_2 y_3 y_4},
\end{aligned}
\tag{89}
$$

where we used the definitions for $n$-point functions and the effective action, the U(1) symmetry, and $\mathbf{y}$ refers to position variables $y_1, .., y_4$. Going to Fourier space $G^{\mathbf{c},(2)}_{xy} = \int_p G^{\mathbf{c},(2)}_p \exp(ip(x-y))$, we get

$$
G^{\mathbf{c},(4)}_{x_1 x_2 x_3 x_4} = -\int_{\mathbf{p}} G^{\mathbf{c},(2)}_{p_1} G^{\mathbf{c},(2)}_{p_2} G^{\mathbf{c},(2)}_{p_3} G^{\mathbf{c},(2)}_{p_4} e^{ip_1 x_1 + ip_2 x_2} e^{-ip_3 x_3 - ip_4 x_4} \Gamma^{(4)}_{p_1 p_2 p_3 p_4},
\tag{90}
$$

such that

$$
G^{\mathbf{c},(4)}_{p_1 p_2 p_3 p_4} = -G^{\mathbf{c},(2)}_{p_1} G^{\mathbf{c},(2)}_{p_2} G^{\mathbf{c},(2)}_{p_3} G^{\mathbf{c},(2)}_{p_4} \Gamma^{(4)}_{p_1 p_2 p_3 p_4}.
\tag{91}
$$

Here we used

$$\Gamma^{(4)}_{p_1 p_2 p_3 p_4} = \int_{\mathbf{y}} e^{-ip_1 y_1 - ip_2 y_2} e^{ip_3 y_3 + ip_4 y_4} \Gamma^{(4)}_{y_1 y_2 y_3 y_4}, \tag{92}$$

$$G^{\mathbf{c},(4)}_{p_1 p_2 p_3 p_4} = \int_{\mathbf{x}} e^{-ip_1 x_1 - ip_2 x_2} e^{ip_3 x_3 + ip_4 x_4} G^{\mathbf{c},(4)}_{x_1 x_2 x_3 x_4}. \tag{93}$$

Using the translation invariance in real-space one finds that $\Gamma^{(4)}_{x_1 x_2 x_3 x_4}$ is independent of the sum of its arguments $x_1 + x_2 + x_3 + x_4$ such that the integration over this component results in a momentum conserving factor $\delta(p_1 + p_2 - p_3 - p_4)$ in Fourier space.

The second and third diagrams are given accordingly by

$$\begin{aligned}
G^{\mathbf{c},(6)}_{p_1 \cdot \cdot p_6} = &-G^{\mathbf{c},(2)}_{p_1} G^{\mathbf{c},(2)}_{p_2} G^{\mathbf{c},(2)}_{p_3} G^{\mathbf{c},(2)}_{p_4} G^{\mathbf{c},(2)}_{p_5} G^{\mathbf{c},(2)}_{p_6} \times \Gamma^{(6)}_{p_1 \cdot \cdot p_6} \\
&+ \int_q G^{\mathbf{c},(2)}_{p_1} G^{\mathbf{c},(2)}_{p_2} G^{\mathbf{c},(2)}_{p_3} \Gamma^{(4)}_{p_1 p_2 p_3 q} G^{\mathbf{c},(2)}_q \Gamma^{(4)}_{q p_4 p_5 p_6} G^{\mathbf{c},(2)}_{p_4} G^{\mathbf{c},(2)}_{p_5} G^{\mathbf{c},(2)}_{p_6} \\
&+ \text{permutations}. \tag{94}
\end{aligned}$$

# B   Loop expressions

The loop diagrams which involve the scattering of two and three particles are summarized in the function $\mathcal{M}$, and diagrammatically displayed in Fig. 2 for $N = 1$. In this section, we derive the analytic expressions which underlie the individual diagrams contributing to the evolution of $\Gamma^{(4)}$. The loop expressions originate from field-derivatives acting on the second line of Eq. (16), where we focus on one representative of each type of diagram (without listing complex conjugates or trivial permutations). Also, the integration over internal indices will be implied throughout this section. We consider

$$\frac{1}{Z_t[J^{(*)}]} \frac{\delta^3 Z_t[J^{(*)}]}{(\delta J_x)^2 \delta J_x^*} \frac{\delta \Gamma_t}{\delta \psi_x^*} = G^{(3)}_{xx,x} \frac{\delta \Gamma_t}{\delta \psi_x^*}, \tag{95}$$

where $G^{(3)}_{xx,x} = \langle (\hat{\psi}_x^\dagger)^2 \hat{\psi}_x \rangle_{J,\text{sym}}$ is non-zero in the presence of a source $J$. The expression may be split into its connected components and is subsequently differentiated with respect to the four external fields:

- **First**, one gets

$$\frac{\delta}{\delta \psi_{x_1}} \frac{\delta}{\delta \psi_{x_2}} \frac{\delta}{\delta \psi_{x_3}^*} \frac{\delta}{\delta \psi_{x_4}^*} \left[ \psi_x^* \psi_x^* \psi_x \frac{\delta \Gamma_t}{\delta \psi_x^*} \right] \supset \delta_{x_1 x_4} \delta_{x_2 x_4} \delta_{x_3 x_4} \Gamma^{(2)}_{x_1 x_4} \to \;\; \text{⨉} \;\;, \tag{96}$$

  representing the bare classical scattering vertex.

- **Secondly**, one finds terms of the kind

$$\frac{\delta}{\delta \psi_{x_1}} \frac{\delta}{\delta \psi_{x_2}} \frac{\delta}{\delta \psi_{x_3}^*} \frac{\delta}{\delta \psi_{x_4}^*} \left[ \psi_x^* G^{\mathbf{c},(2)}_{xx} \frac{\delta \Gamma_t}{\delta \psi_x^*} \right], \tag{97}$$

$$\frac{\delta}{\delta \psi_{x_1}} \frac{\delta}{\delta \psi_{x_2}} \frac{\delta}{\delta \psi_{x_3}^*} \frac{\delta}{\delta \psi_{x_4}^*} \left[ \psi_x \tilde{G}^{\mathbf{c},(2)}_{xx} \frac{\delta \Gamma_t}{\delta \psi_x^*} \right], \tag{98}$$

with $\tilde{G}^{\mathbf{c},(2)}_{xx} = \langle(\hat{\psi}^*_x)^2\rangle^{\mathbf{c}}_J$. Applying all derivatives in (97) to the first and the last factor in the bracket, we get the first diagram of $\mathcal{M}$

$$\frac{\delta}{\delta\psi_{x_1}}\frac{\delta}{\delta\psi_{x_2}}\frac{\delta}{\delta\psi^*_{x_3}}\frac{\delta}{\delta\psi^*_{x_4}}\left[\psi^*_x G^{\mathbf{c},(2)}_{xx}\frac{\delta\Gamma_t}{\delta\psi^*_x}\right] \supset -G^{\mathbf{c},(2)}_{x_1x_1}\Gamma^{(4)}_{x_1x_2x_3x_4} \rightarrow \quad . \tag{99}$$

For translation invariant systems this diagram cancels against its permutations. The analogous expression (98) vanishes as $\tilde{G}^{\mathbf{c},(2)}_{xx} = 0$ when setting $J = 0$. When acting two derivatives on the two-point function in (97), we get

$$\frac{\delta}{\delta\psi_{x_1}}\frac{\delta}{\delta\psi_{x_2}}\frac{\delta}{\delta\psi^*_{x_3}}\frac{\delta}{\delta\psi^*_{x_4}}\left[\psi^*_x G^{\mathbf{c},(2)}_{xx}\frac{\delta\Gamma_t}{\delta\psi^*_x}\right] \supset -\Gamma^{(4)}_{x_1y_1y_2x_4}G^{\mathbf{c},(2)}_{x_3y_1}G^{\mathbf{c},(2)}_{y_2x_3}\Gamma^{(2)}_{x_3x_2} \rightarrow \quad , \tag{100}$$

by using that

$$\frac{\delta}{\delta\psi_{x_1}}\frac{\delta}{\delta\psi^*_{x_2}}G^{\mathbf{c},(2)}_{xz}\bigg|_{\psi^{(*)}=0} = -\Gamma^{(4)}_{x_1y_1y_2x_2}G^{\mathbf{c},(2)}_{xy_1}G^{\mathbf{c},(2)}_{y_2z}, \tag{101}$$

where contributions involving the three-vertex vanish due to U(1) invariance. Similarly, an analogous contribution arises from (97)

$$\frac{\delta}{\delta\psi_{x_1}}\frac{\delta}{\delta\psi_{x_2}}\frac{\delta}{\delta\psi^*_{x_3}}\frac{\delta}{\delta\psi^*_{x_4}}\left[\psi^*_x G^{\mathbf{c},(2)}_{xx}\frac{\delta\Gamma_t}{\delta\psi^*_x}\right] \supset -\Gamma^{(4)}_{x_1y_1y_2x_4}G^{\mathbf{c},(2)}_{x_3y_1}G^{\mathbf{c},(2)}_{y_2x_3}\Gamma^{(2)}_{x_3x_2} \rightarrow \quad . \tag{102}$$

- **Third**, we consider

$$\frac{\delta}{\delta\psi_{x_1}}\frac{\delta}{\delta\psi_{x_2}}\frac{\delta}{\delta\psi^*_{x_3}}\frac{\delta}{\delta\psi^*_{x_4}}\left[G^{\mathbf{c},(3)}_{xx,x}\frac{\delta\Gamma_t}{\delta\psi^*_x}\right]. \tag{103}$$

By U(1) symmetry, the only surviving contributions are obtained by applying an odd number of derivatives to each factor inside the brackets.

For one derivative acting on the three-point function, we use that

$$\frac{\delta}{\delta\psi^*_{x_4}}G^{\mathbf{c},(3)}_{xx,x}\bigg|_{\psi^{(*)}=0} = -\Gamma^{(4)}_{x_4z_1z_2z_3}G^{\mathbf{c},(2)}_{xz_1}G^{\mathbf{c},(2)}_{z_2x}G^{\mathbf{c},(2)}_{z_3x} \rightarrow \quad , \tag{104}$$

which yields the diagram in combination with the other derivatives acting on the second factor to yield $\Gamma^{(4)}_{xx_3x_1x_2}$.

In the case where three derivatives act on the three-point function, we get terms of the kind

$$\frac{\delta}{\delta\psi_{x_2}}\frac{\delta}{\delta\psi^*_{x_3}}\frac{\delta}{\delta\psi^*_{x_4}}G^{\mathbf{c},(3)}_{xx,x}\bigg|_{\psi^{(*)}=0} \rightarrow -\Gamma^{(6)}_{x_3x_4z_1x_2z_2z_3}G^{\mathbf{c},(2)}_{xz_1}G^{\mathbf{c},(2)}_{z_2x}G^{\mathbf{c},(2)}_{z_3x} \rightarrow \quad , \tag{105}$$

where we again augmented the result with the second factor, here $\Gamma^{(2)}_{xx_1}$, or alternatively

$$\begin{aligned}
\frac{\delta}{\delta\psi_{x_2}}\frac{\delta}{\delta\psi^*_{x_3}}\frac{\delta}{\delta\psi^*_{x_4}}G^{\mathbf{c},(3)}_{xx,x}\bigg|_{\psi^{(*)}=0} &= \frac{\delta}{\delta\psi_{x_2}}\frac{\delta}{\delta\psi^*_{x_3}}\left[-\Gamma^{(4)}_{x_4z_1z_2z_3}G^{\mathbf{c},(2)}_{xz_1}G^{\mathbf{c},(2)}_{z_2x}G^{\mathbf{c},(2)}_{z_3x}\right] \\
&\supset -\Gamma^{(4)}_{x_4z_1z_2z_3}G^{\mathbf{c},(2)}_{xz_1}G^{\mathbf{c},(2)}_{z_2x}\frac{\delta}{\delta\psi_{x_2}}\frac{\delta}{\delta\psi^*_{x_3}}\left[G^{\mathbf{c},(2)}_{z_3x}\right] \\
&= \Gamma^{(4)}_{x_4z_1z_2z_3}G^{\mathbf{c},(2)}_{xz_1}G^{\mathbf{c},(2)}_{z_2x}\Gamma^{(4)}_{x_3y_1y_2x_2}G^{\mathbf{c},(2)}_{z_3y_1}G^{\mathbf{c},(2)}_{y_2x} \\
&\rightarrow \quad , \tag{106}
\end{aligned}$$

including again $\Gamma^{(2)}_{xx_1}$.

Analogously, we give the expression for the computation of the conjugate of the diagram shown in Eq. (100). This diagram will be especially important for the computation of observables in the $1/N$ expansion of the effective action. It originates from the term

$$-\frac{1}{Z_t[J^{(*)}]}\frac{\delta^3 Z_t[J^{(*)}]}{(\delta J^*_x)^2\delta J_x}\frac{\delta\Gamma_t}{\delta\psi_x} = -G^{(3)}_{x,xx}\frac{\delta\Gamma_t}{\delta\psi_x}, \tag{107}$$

where $G^{(3)}_{x,xx} = 1/2\langle\{\hat\psi^\dagger_x,\hat\psi^2_x\}\rangle_J$. Again applying the external derivatives, we get

$$-\frac{\delta}{\delta\psi_{x_1}}\frac{\delta}{\delta\psi_{x_2}}\frac{\delta}{\delta\psi^*_{x_3}}\frac{\delta}{\delta\psi^*_{x_4}}\left[\psi_x G^{\mathbf{c},(2)}_{xx}\frac{\delta\Gamma_t}{\delta\psi_x}\right] \supset -\Gamma^{(4)}_{x_1 y_1 y_2 x_4}G^{\mathbf{c},(2)}_{x_2 y_1}G^{\mathbf{c},(2)}_{y_2 x_2}\Gamma^{(2)}_{x_2 x_3} \to \ \raisebox{-0.6em}{\includegraphics[height=1.2em]{diag108}} \ . \tag{108}$$

## C  $1/N$ **counting of diagrams**

In this section, we briefly review the counting of powers of $N$ in loop diagrams. While the Hamiltonian is invariant under a global U($N$) symmetry, there are $N$ U(1) subgroups which imply the conservation of particle number for the individual components $i$. Since we diagonalized the two-point function in field space, i.e. $G^{\mathbf{c},(2)}_{\alpha_1\alpha_2} = G^{\mathbf{c},(2)}_{x_1 x_2}\delta_{i_1 i_2}$, lines carrying a certain field index are never interrupted throughout a diagram. For the effective vertices, we get

$$\Gamma^{(4)}_{\alpha_1..\alpha_4} \propto \delta_{i_1 i_3}\delta_{i_2 i_4} + \text{perm.}, \tag{109}$$

$$\Gamma^{(6)}_{\alpha_1..\alpha_6} \propto \delta_{i_1 i_4}\delta_{i_2 i_5}\delta_{i_3 i_6} + \text{perm.}, \tag{110}$$

and analogously for all higher-order vertices.

Depending on field indices and orientation of vertices, we can differentiate between different contributions, illustrated here for the example of a one-loop diagram

$$\raisebox{-0.9em}{\includegraphics[height=2em]{diag111}} = \mathcal{O}(1/N^2), \tag{111}$$

$$\raisebox{-0.9em}{\includegraphics[height=2em]{diag112}} = \mathcal{O}(1/N^2)\times\mathcal{O}(N), \tag{112}$$

where red lines follow field components through the diagrams. The factor of $1/N^2$ originates from the two bare vertices, which come with a factor of $1/N$ each. The additional factor of $N$ in the second diagram originates from the summation over all possible intermediate particles within the loop. In general, each such closed loop of field indices yields a factor of $N$. Using this and assuming Eq. (45), we get the following power counting for the diagrams shown in Fig. 2 (where we always pick the representative with the largest power of $N$). Subleading

diagrams of Fig. 2 are the following

$$\text{(diagram)} = \mathcal{O}(1/N^2), \tag{113}$$

$$\text{(diagram)} = \mathcal{O}(1/N^3) \times N = \mathcal{O}(1/N^2), \tag{114}$$

$$\text{(diagram)} = \mathcal{O}(1/N^3) \times N = \mathcal{O}(1/N^2), \tag{115}$$

$$\text{(diagram)} = \mathcal{O}(1/N^3) \times N = \mathcal{O}(1/N^2). \tag{116}$$

The dominant contribution at order $1/N$ is given by

$$\text{(diagram)} = \mathcal{O}(1/N^2) \times N = \mathcal{O}(1/N). \tag{117}$$

Eventually, we can confirm self-consistently that all contributions lead to an evolution of the four-vertex at order $\mathcal{O}(1/N)$. As $\Gamma^{(4)}$ is initially sourced by the bare vertex at order $\mathcal{O}(1/N)$, it confirms our initial assumption Eq. (45). This line of argument can be applied to the entire hierarchy of equal-time vertices: $\Gamma^{(6)}$ is sourced by a combination of the bare vertex and $\Gamma^{(4)}$ (i.e. at order $\mathcal{O}(1/N^2)$) and the corresponding diagrams for its evolution remain at this order.

## D  Proof of Eq. (71)

Here, we explicitly demonstrate that the expression (71) is a solution to the imaginary part of Eq. 70. First, we make use of the defining property of $\Gamma^B$, to define

$$\Gamma^B_{pqrs} \equiv (\Gamma_p - \Gamma_s)(\Gamma_q - \Gamma_r) B_{pqrs}, \tag{118}$$

where we made the external inverse propagators explicit. With this, Eq. (71) becomes

$$B_{pqrs} = \frac{g^2/(4N)}{-\Delta\omega_{pqrs} + i\epsilon} \int_{k'} \left( -\frac{G_{k'}G_{k''} - \frac{1}{4}}{\Delta\omega_{k'psk''} - i\epsilon} \frac{1}{1 + \Pi_{k'k''}} \underbrace{- \frac{G_{k'}G_{k''} - \frac{1}{4}}{\Delta\omega_{k''qrk'} - i\epsilon} \frac{1}{1 + \Pi_{k''k'}}}_{(I)} \right)$$

$$\underbrace{- \frac{g/2}{-\Delta\omega_{pqrs} + i\epsilon} \int_{k'} (G_{k''} - G_{k'}) B_{pk'k''s}}_{(II)} \underbrace{- \frac{g/2}{-\Delta\omega_{pqrs} + i\epsilon} \int_{k'} (G_{k'} - G_{k''}) B_{k''qrk'}}_{(III)}, \tag{119}$$

and our ansatz for the solution (71) is accordingly given by

$$\text{Im}(B_{pqrs}) = \mathcal{P}\left[ \frac{g g^{\text{eff}}_{ps}/(4N)}{-\Delta\omega_{pqrs} + i\epsilon} \right] \int_{k'} \pi\delta(\Delta\omega_{pk'k''s})\left( G_{k'}G_{k''} - \frac{1}{4} \right) + \{p,s \leftrightarrow q,r\}. \tag{120}$$

In the following, we will strip Eq. (119) into its individual parts and analyse terms separately.

### D.1 First term: (I)

Considering the first term of Eq. (119), we use the symmetry relation (135) to get

$$\delta(\Delta\omega_{pqrs})\int_{k'}\mathrm{Re}\left(-\frac{G_{k'}G_{k''}-\frac{1}{4}}{\Delta\omega_{k'psk''}-i\epsilon}\frac{1}{1+\Pi_{k'k''}}-\frac{G_{k'}G_{k''}-\frac{1}{4}}{\Delta\omega_{k''qrk'}-i\epsilon}\frac{1}{1+\Pi_{k''k'}}\right)=0\,. \qquad (121)$$

Using this, the imaginary part of the first term in Eq. (119) is given by

$$\mathrm{Im}(I)=\mathcal{P}\left(\frac{g^2/(4N)}{-\Delta\omega_{pqrs}+i\epsilon}\right)\times\mathrm{Im}\left[\int_{k'}\frac{G_{k'}G_{k''}-\frac{1}{4}}{\Delta\omega_{k'psk''}-i\epsilon}\frac{1}{1+\Pi_{k'k''}}+\{p,q\leftrightarrow r,s\}\right]. \qquad (122)$$

Next, we compute the second factor separately,

$$\int_{k'}\mathrm{Im}\left(\frac{G_{k'}G_{k''}-\frac{1}{4}}{\Delta\omega_{k'psk''}-i\epsilon}\frac{1}{1+\Pi_{k'k''}}\right)=\int_{k'}\pi\delta(\Delta\omega_{k'psk''})\left(G_{k'}G_{k''}-\frac{1}{4}\right)\frac{1+\mathrm{Re}(\Pi^*_{k'k''})}{|1+\Pi_{k'k''}|^2}$$

$$+\int_{k'}\mathrm{Re}\left(\frac{G_{k'}G_{k''}-\frac{1}{4}}{\Delta\omega_{k'psk''}-i\epsilon}\right)\frac{\mathrm{Im}(\Pi^*_{k'k''})}{|1+\Pi_{k'k''}|^2}\,, \qquad (123)$$

### D.2 Second and third term: (II) and (III)

Similarly, for the second and third term of Eq. (119), which involves $B$ itself, we have

$$\mathrm{Im}\left[\frac{g/2}{-\Delta\omega_{pqrs}+i\epsilon}\right]\int_{k'}(G_{k''}-G_{k'})\mathrm{Re}\left[B_{pk'k''s}-B_{k''qrk'}\right]$$

$$=-\pi\frac{g}{2}\delta(\Delta\omega_{pqrs})\int_{k'}(G_{k''}-G_{k'})\mathrm{Re}\left[B_{pk'k''s}-B^*_{pk'k''s}\right]$$

$$=0\,, \qquad (124)$$

where we used the identity $B^*_{pk'k''s}=B_{sk''k'p}$, see section F. As a consequence, we focus on the following combination of real and imaginary parts

$$-\frac{g}{2}\mathrm{Re}\left[\frac{1}{-\Delta\omega_{pqrs}+i\epsilon}\right]\int_{k'}(G_{k''}-G_{k'})\mathrm{Im}(B_{pk'k''s})=-\mathcal{P}\left[\frac{g/2}{-\Delta\omega_{pqrs}+i\epsilon}\right]\int_{k'}(G_{k''}-G_{k'})\mathrm{Im}(B_{pk'k''s})\,. \qquad (125)$$

Next, we compute the second factor of this term by employing the definition of $B$, i.e. Eq (120)

$$\int_{k'}(G_{k''}-G_{k'})\mathrm{Im}(B_{pk'k''s})=\int_{q',k'}\left(G_{q'}G_{q''}-\frac{1}{4}\right)\left[\frac{gg^{\mathrm{eff}}_{ps}}{4}\mathcal{P}\left[\frac{(G_{k''}-G_{k'})}{-\Delta\omega_{pk'k''s}+i\epsilon}\right]\pi\delta(\Delta\omega_{pq'q''s})\right.$$

$$+\left.\frac{gg^{\mathrm{eff}}_{k'k''}}{4}\mathcal{P}\left[\frac{(G_{k''}-G_{k'})}{-\Delta\omega_{pk'k''s}+i\epsilon}\right]\pi\delta(\Delta\omega_{k'q''q'k''})\right]$$

$$=\frac{gg^{\mathrm{eff}}_{ps}}{4}\mathcal{P}(\Pi_{ps})\int_{q'}\pi\delta(\Delta\omega_{pq'q''s})\left(G_{q'}G_{q''}-\frac{1}{4}\right)\int_{q'}\frac{gg^{\mathrm{eff}}_{q'q''}}{4}\mathcal{P}\left[\frac{G_{q'}G_{q''}-\frac{1}{4}}{-\Delta\omega_{pq'q''s}+i\epsilon}\right]\mathrm{Im}(\Pi_{q''q'})\,. \qquad (126)$$

Putting all terms together, we observe

$$\mathrm{Im(I)}+\mathrm{Im(II)}+\mathrm{Im(III)}=\mathcal{P}\left(\frac{g^2/(4N)}{-\Delta\omega_{pqrs}+i\epsilon}\right)\int_{k'}\pi\delta(\Delta\omega_{k'psk''})\frac{G_{k'}G_{k''}-\frac{1}{4}}{|1+\Pi_{k'k''}|^2}+\{p,s\leftrightarrow q,r\}$$

$$=\mathcal{P}\left(\frac{gg^{\mathrm{eff}}_{ps}/(4N)}{-\Delta\omega_{pqrs}+i\epsilon}\right)\int_{k'}\pi\delta(\Delta\omega_{k'psk''})\left(G_{k'}G_{k''}-\frac{1}{4}\right)+\{p,s\leftrightarrow q,r\}$$

$$=\mathrm{Im}(B_{pqrs})\,, \qquad (127)$$

which confirms that the ansatz Eq (120) solves the imaginary part of Eq. (119).

# E  Non-perturbative Boltzmann equation

To derive the non-perturbative Boltzmann equation, we compute

$$\partial_t G_p = -\int_{q,r,s} g\,\mathrm{Im}(\Gamma^A_{pqrs} + \Gamma^B_{pqrs})G_p G_q G_r G_s\,. \tag{128}$$

To this end, we again split the expression into individual parts

## E.1  $\mathrm{Im}(\Gamma^A)$

We first consider the following term, see also (66),

$$\int_{q'} \mathrm{Im}\left(\Gamma^A_{pq'q''s}\right)G_{q'}G_{q''} = -\int_{q'} \pi G_{q'}G_{q''}\delta(\Delta\omega_{pq'q''s})V^{\mathrm{eff}}_{pq'q''s}$$
$$-\int_{q'} \pi G_{q'}G_{q''}\delta(\Delta\omega_{pq'q''s})V^{\mathrm{eff}}_{pq'q''s}\mathcal{P}(\Pi_{ps})$$
$$+\int_{q'} \mathcal{P}\left[\frac{G_{q'}G_{q''}}{\Delta\omega_{pq'q''s}-i\epsilon}\right]\left[V^{\mathrm{eff}}_{p\underline{q'q''}s}\mathrm{Im}\left(\Pi_{ps}\right) + \{p,s \leftrightarrow q',q''\}\right], \tag{129}$$

where we adapted the notation of Eqs. (62) and (63). Here, the second line is given by

$$-\int_{q'} \pi G_{q'}G_{q''}\delta(\Delta\omega_{pq'q''s})V^{\mathrm{eff}}_{pq'q''s}\mathcal{P}(\Pi_{ps}) = -(\Gamma_p - \Gamma_s)\frac{g^{\mathrm{eff}}_{ps}}{2N}\mathcal{P}(\Pi_{ps})\int_{q'} \pi\delta(\Delta\omega_{pq'q''s})\left(G_{q'}G_{q''} - \frac{1}{4}\right)$$
$$+\frac{g^{\mathrm{eff}}_{ps}}{2N}\mathcal{P}(\Pi_{ps})\mathrm{Im}(\Pi_{ps})\left(1 - \frac{\Gamma_p\Gamma_s}{4}\right). \tag{130}$$

The third line can be rewritten to yield

$$\int_{q'} \mathcal{P}\left[\frac{G_{q'}G_{q''}}{\Delta\omega_{pq'q''s}-i\epsilon}\right]V^{\mathrm{eff}}_{p\underline{q'q''}s}\mathrm{Im}\left(\Pi_{ps}\right) = -\frac{g^{\mathrm{eff}}_{ps}}{2N}\mathcal{P}(\Pi_{ps})\mathrm{Im}(\Pi_{ps})\left(1 - \frac{\Gamma_p\Gamma_s}{4}\right), \tag{131}$$

and similarly, we obtain for the fourth line

$$\int_{q'} \mathcal{P}\left[\frac{G_{q'}G_{q''}}{\Delta\omega_{pq'q''s}-i\epsilon}\right]V_{q'\underline{ps}q''}\frac{\mathrm{Im}\left(\Pi_{q'q''}\right)}{|1+\Pi_{q'q''}|^2} = (\Gamma_p - \Gamma_s)\int_{q',k'} \frac{g\,g^{\mathrm{eff}}_{k'k''}}{4N}\left(G_{q'}G_{q''} - \frac{1}{4}\right)\pi \tag{132}$$
$$\times \delta(\Delta\omega_{q'k''k'q''})\mathcal{P}\left[\frac{G_{k'} - G_{k''}}{\Delta\omega_{pk'k''s}-i\epsilon}\right].$$

## E.2  $\mathrm{Im}(\Gamma^B)$

The second term can be brought into the form

$$\int_{q'} \mathrm{Im}\left(\Gamma^B_{pq'q''s}\right)G_{q'}G_{q''} = -(\Gamma_p - \Gamma_s)\frac{g^{\mathrm{eff}}_{ps}}{2N}\mathcal{P}(\Pi_{ps})\int_{q'} \pi\delta(\Delta\omega_{pk'k''s})\left(G_{k'}G_{k''} - \frac{1}{4}\right)$$
$$-(\Gamma_p - \Gamma_s)\int_{q',k'} \frac{g\,g^{\mathrm{eff}}_{q'q''}}{4N}\left(G_{k'}G_{k''} - \frac{1}{4}\right)\pi\delta(\Delta\omega_{q'k''k'q''})\mathcal{P}\left[\frac{G_{q'} - G_{q''}}{\Delta\omega_{pq'q''s}-i\epsilon}\right]. \tag{133}$$

In summary, all but one term cancel, and we get

$$\int_{q'} \mathrm{Im}\left(\Gamma^A_{pq'q''s} + \Gamma^B_{pq'q''s}\right)G_{q'}G_{q''} = -\int_{q'} \pi G_{q'}G_{q''}\delta(\Delta\omega_{pq'q''s})V^{\mathrm{eff}}_{pq'q''s}\,, \tag{134}$$

which straightforwardly leads to Eq. (73).

## F  Symmetries under coordinate exchange

In our calculations we frequently use the symmetry of certain objects under the momentum exchange $p, s \leftrightarrow s, p$ (i.e. $p, s \leftrightarrow q, r$ in the presence of the momentum and energy conserving delta functions). For instance, we get for the one-loop self-energy function

$$\Pi_{sp} = g \int_{k'} \frac{G_{k'} - G_{k'-(p-s)}}{\Delta\omega_{sk'(k'-(p-s))p} - i\epsilon} = g \int_{k'} \frac{G_{k'+(p-s)} - G_{k'}}{\Delta\omega_{s(k'+(p-s))k'p} - i\epsilon}$$
$$= g \int_{k'} \frac{G_{k'} - G_{k'+(p-s)}}{\Delta\omega_{pk'(k'+(p-s))s} + i\epsilon} = \Pi_{ps}^* \,. \tag{135}$$

Here, a crucial ingredient was the anti-symmetry of the numerator under the coordinate flip.

We employ similar arguments to demonstrate the relation $B_{pqrs}^* = B_{rspq}$ which we use to solve the non-perturbative evolution equation for $\mathrm{Im}(\Gamma^B)$. First, we show the identity at the one-loop level, cf. Eq. (68). Using

 $\propto \dfrac{1}{N} \dfrac{\Gamma_r \Gamma_s}{\Delta\omega_{pqrs} - i\epsilon} \displaystyle\int_{k'} \dfrac{G_{k'} G_{k''} - \frac{1}{4}}{\Delta\omega_{pk'k''s} - i\epsilon} \,. \tag{136}$

one gets

$$B_{pqrs} = \frac{-g^2/N}{\Delta\omega_{pqrs} - i\epsilon} \int_{k'} \frac{G_{k'} G_{k''} - \frac{1}{4}}{\Delta\omega_{pk'k''s} - i\epsilon} + \text{higher-oder loops} \,. \tag{137}$$

In presence of the energy and momentum conserving $\delta$-distributions $\delta(\Delta\omega_{pqrs})\delta(p+q-r-s)$, we find

$$\left[ \frac{-g^2/N}{\Delta\omega_{pqrs} - i\epsilon} \int_{k'} \frac{G_{k'} G_{k''} - \frac{1}{4}}{\Delta\omega_{pk'k''s} - i\epsilon} \right]^* = \frac{-g^2/N}{\Delta\omega_{pqrs} + i\epsilon} \int_{k'} \frac{G_{k'} G_{k''} - \frac{1}{4}}{\Delta\omega_{pk'k''s} + i\epsilon}$$
$$= \frac{-g^2/N}{\Delta\omega_{rspq} - i\epsilon} \int_{k'} \frac{G_{k'} G_{k''} - \frac{1}{4}}{\Delta\omega_{k''spk'} - i\epsilon} = \frac{-g^2/N}{\Delta\omega_{rspq} - i\epsilon} \int_{k'} \frac{G_{k'} G_{k'+(p-s)} - \frac{1}{4}}{\Delta\omega_{(k'+(p-s))spk'} - i\epsilon}$$
$$= \frac{-g^2/N}{\Delta\omega_{rspq} - i\epsilon} \int_{k'} \frac{G_{k'+(s-p)} G_{k'} - \frac{1}{4}}{\Delta\omega_{sk'(k'+(s-p))p} - i\epsilon} = \left[ \frac{-g^2/N}{\Delta\omega_{pqrs} - i\epsilon} \int_{k'} \frac{G_{k'} G_{k''} - \frac{1}{4}}{\Delta\omega_{pk'k''s} - i\epsilon} \right]_{p,q \leftrightarrow r,s} \,. \tag{138}$$

With this, the identity $B_{pqrs}^* = B_{rspq}$ is easily shown at the one-loop level. The generalization to any loop order, and thus the full expression $B$, follows in the same manner iteratively from Eq. (119), where it is respected in each term individually.

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
