# Peer review of "Equal-time approach to real-time dynamics of quantum fields"

_SciPost Physics, doi:SciPost Phys. 14, 011 (2023)_

## Round 1 · Referee Report · Anonymous (Referee 1) · 2022-6-22

Strengths

  1. This is a very detailed well-structured paper written in a logical way.
  2. It contains important and necessary details to help readers to go through the derivations.
  3. The notations are all set and explained in the Section II.
  4. The key equations are illustrated with the Feynman diagrams.
  5. General theory is illustrated with several examples including both perturbative and non-perturbative (large N) approaches.
  6. The Section VI is devoted to the measurement protocol to be implemented with the cold-atom simulators.
  7. Appendices provide sufficient details for the experts.

Report

This is a very interesting technical paper addressing general questions of the quantum field theoretical approach to the out-of equilibrium quantum systems. Besides, it addresses the measurement protocols to be implemented with cold-atom quantum simulators. My personal opinion is that the paper will attract an attention of different target groups from PhD students who learn many-body techniques to the experts actively working in the field. I think that the paper is well written and would not suggest any changes at present stage.

---

## Round 1 · Referee Report · Anonymous (Referee 2) · 2022-8-15

Report

Usually (e.g., such as expounded in Calzetta and Hu, Nonequilibrium Quantum Field Theory, CUP (2008)) nonequilibrium quantum fields are described in terms of unequal time correlations. This raises some issues of principle (for example, whether causality is enforced when deriving kinetic theory from quantum field theory) and most importantly does not match actual experimental practice. This paper shows that it is possible to develop nonequilibrium quantum field theory completely in terms of equal time correlations. The paper is highly formal - essentially it translates to the language of equal time correlations everything one already knows about the unequal time ones- but opens up new avenues in nonequilibrium quantum field theory and will be useful to the community.

---

## Editorial Decision

published